# FROM SHORTCUTS TO REASONING: ROBUST POST-TRAINING OF THEORY OF MIND WITH REINFORCEMENT LEARNING

## ABSTRACT

Theory of Mind (ToM) is a must-acquire skill for modern foundation model systems to operate effectively and safely in the real world. Recent works have explored honing ToM via post-training; however, we show that such progress is confounded by a pervasive "shortcut" issue: tasks can reach up to *99%* accuracy by simply exploiting *spurious causal correlations*, leading to a false sense of ToM. Motivated by this, we first develop a framework to systematically examine ToM datasets for shortcuts and provide guidance for future development. We find that questions reducible to *pure state tracking* (e.g. "belief") are especially shortcut-prone compared to mind questions (e.g. "intention") where *reasoning beyond tracking* is required. Using four shortcut-free datasets across three ToM contexts, we then comprehensively study whether Reinforcement Fine-Tuning with verifiable rewards and explicit reasoning chains (Thinking-RFT) elevates ToM beyond Supervised Fine-Tuning (SFT). Our key findings are: 1) Thinking-RFT effectively improves ToM in all scenarios (*+6%* vs. SFT), particularly in complex higher-order reasoning (*+10%* vs. SFT) and multimodal cases (*+7%* vs. SFT), and generalizes notably better to unseen domains and higher-order queries while being more robust to counterfactuals. 2) ToM benefits specifically from the *joint effect of reasoning and RL*: Thinking-RFT outperforms No-Thinking-RFT by 7% on average. 3) RFT works by learning to ground its reasoning on anchor cues (keywords/state changes) that correspond to causal factors. We believe our study is useful for developing effective and robust ToM post-training datasets and advancing critical ToM capabilities in foundation models.

## 1 INTRODUCTION

Theory of Mind (ToM), the capacity to reason about and infer latent mental states such as beliefs, desires, intentions, and knowledge of other people, is fundamental in human social cognition (Dennett, 1988; Gopnik & Wellman, 2012). Equipping modern AI systems with ToM-like reasoning is key for natural, safe, and personalized interaction, moving it closer to genuine human-like intelligence.

Recent work has shown that current foundation models still lack human-level ToM reasoning skills (He et al., 2023; Sclar et al., 2025; Jin et al., 2024; Shi et al., 2025), and has proposed fixes by guiding the model through carefully structured reasoning procedures using targeted prompts (Wilf et al., 2024), agentic frameworks (Zhang et al., 2025b; Shi et al., 2025), or Bayesian inference over mental-state models (Jin et al., 2024; Zhang et al., 2025a). However, these approaches typically rely on explicitly designed multi-step prompting around a strong backbone (e.g., GPT-4o), which increases inference-time complexity and makes deployment more challenging.

In this paper, we instead ask whether we can *directly teach* the base model such advanced ToM skills via post-training. We focus on Reinforcement Learning with Verifiable Rewards (RLVR) (Shao et al., 2024), which encourages the model to produce explicit chains of thought whose final answers can be checked by simple rules. RLVR-style Fine-Tuning with elicit reasoning chains (hereafter referred to as "Thinking-RFT") has recently proven highly effective at incentivizing reasoning in math and logic-heavy tasks (Guo et al., 2025; Kimi et al., 2025; Kumar et al., 2025; Yu et al., 2025). We conduct a ToM-tailored empirical study of post-training with Supervised Fine-Tuning

(SFT), Thinking-RFT, and No-Thinking-RFT (RLVR without reasoning chains) on shortcut-free ToM benchmarks, and provide an in-depth analysis of when and how explicit reasoning helps ToM.

However, when we initially attempted to identify the best ToM post-training strategy on standard benchmarks, we encountered puzzling behaviors that led us to suspect that many ToM datasets can be "solved" via *shortcuts* rather than genuine mental-state reasoning. Specifically, we began our exploration by post-training on popular ToM datasets FANToM (Kim et al., 2023) and Hi-ToM (He et al., 2023), both widely used in prior post-training studies (Kim et al., 2023; Lu et al., 2025; Sarangi & Salam, 2025). Surprisingly, on Hi-ToM we found that *higher-order* queries were even easier than *lower-order* ones (3rd/4th-order >95% vs. 1st/2nd at 89.5%), and the trained model produced incoherent reasoning traces (Figure 1). A simple manual check revealed a near-perfect shortcut: solution is the object location when the outermost agent leaves the scene, regardless.

This observation motivates us to first systematically examine the landscape of ToM datasets. We devise a simple auditing framework combining AI exploration scores and algebraic lexical similarity, and find that four out of eight popular benchmarks suffer from severe shortcut issues. This is critical as experiments confirm that 1) models trained on shortcut datasets fail to provide *logically sound* reasoning traces 90% of the time while exhibiting significant loss in generalization capability, and 2) it produces misleading performance numbers that prevent us from faithfully identifying the actual best post-training strategy. These drawbacks could potentially misguide both practitioners and future research in the ToM community.

We then focus our post-training study on shortcut-free datasets spanning conversational, narrative, and multimodal ToM. Across these settings, we find that Thinking-RFT 1) successfully learns ToM-specific reasoning strategy and achieves state-of-the-art results on all settings (at most >30% improvement over zero-shot, > 10% over SFT) and 2) shows robust generalization. In other words, it is effective in incentivizing intricate ToM-reasoning skills that were only achievable through rather complex inference-time algorithms previously. Through attention visualizations, we show that Thinking-RFT achieves this by teaching the model to identify and track the key tokens and state changes, enabling more robust and precise reasoning. In summary, our contributions are threefold:

- We raise the *shortcut* issue widely exists in popular ToM datasets. While they are fine for evaluation, we show that they are detrimental to the genuine reasoning incentive. We also develop a simple framework combining AI score and lexical similarity to quantify shortcuts and make useful recommendations for future ToM tuning set making.

- We conduct a comprehensive study of post-training in ToM, considering RFT, SFT, and zero-shot methods, along with detailed experiments on generalization and robustness. We show that RFT successfully elicits ToM in all settings while being generalizable and robust.

- We provide in-depth analysis of the role of reasoning and RL in improving ToM as to why RFT works well. We underscore that reasoning is a crucial part of ToM and for RFT to work on ToM. We furthur highlight that RFT works well by teaching the model to identify and track key information, thus performing precise recursive reasoning.

## 2 THEORY OF MIND: SHORTCUTS OR REASONING?

### 2.1 AUDITING ToM DATASETS FOR SHORTCUTS

We propose a simple framework consisting of two parts that are cheap to compute: 1) *LLM/Agent-guided rules* to identify causal and procedural shortcuts, and 2) *lexical associations* to check for spurious lexical associations; we treat datasets violating either one as shortcut-prone. Crucially, this not only ensures our latter experimental results are sound but also serves as guidance for future ToM tuning set development. More details for exact reproducibility are provided in Appendix E.

**LLM/Agent-guided rules.** Ironically, the frustratingly simple approach of simply asking an advanced LLM or agent to discover potential shortcuts works very well. In fact, our finding suggests that LLMs find 80% of the cheap shortcuts, especially the casual or procedural shortcuts. Concretely, on a small, stratified seed set $\mathcal{D}_{\text{seed}}$, we prompt a frozen LLM to enumerate *candidate shortcut rules* (e.g., *last-seen location*, *world-state leak for belief questions*, *first-mentioned agent*, *option length/position*). Each rule is implemented as a zero-update heuristic $h_k$ and evaluated on a held-out

Table 1: Results of ToM datasets and shortcut (SC) probing.

| ToM Datasets | Format | Vision | Tracking | Intention | SC (Causal) | SC (Lexical) |
|---|---|---|---|---|---|---|
| ExploreToM | narrative | | ✓ | | ✓ | |
| FANToM | conversational | | ✓ | | | ✓ |
| ToMi | narrative | | ✓ | | ✓ | ✓ |
| Hi-ToM | narrative | | ✓ | | ✓ | ✓ |
| OpenToM | narrative | | ✓ | ✓ | | |
| ToMATO | conversational | | ✓ | ✓ | | |
| MMToM | narrative | ✓ | ✓ | ✓ | | |
| MuMA-ToM | narrative | ✓ | ✓ | ✓ | | |

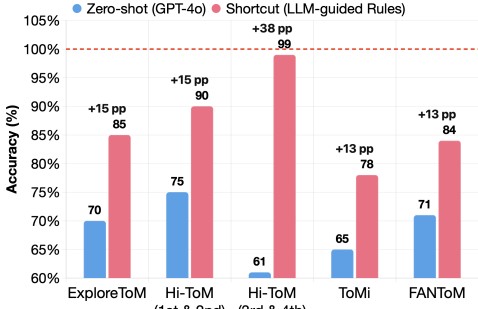

Let's break down the problem step by step: 1. Initially, the beans were in the green_crate. 2. ... 3. ...4. ... Since Jack stayed in the sunroom after the beans were moved to the red_envelope, he would think that the beans are still in the red_envelope. Therefore, Nathan thinks that Emma thinks Jack thinks that the beans are in the red_envelope.

Figure 1: Mis-reasoning.

slice $\mathcal{D}_{\text{probe}}$:

$$A(h_k) = \frac{1}{|\mathcal{D}_{\text{probe}}|} \sum_{(x,y) \in \mathcal{D}_{\text{probe}}} \mathbf{1}\{h_k(x) = y\}.$$

Let $A_0 = \max\{1/K, A_{\text{majority}}\}$ be the random/majority baseline. We accept $h_k$ as a *plausible shortcut* if

$$A(h_k) - A_0 \geq \delta_{\text{abs}}$$

with $\delta_{\text{abs}}$=0.2 by default. We also enforce $p$-value $< 0.05$ for statistical reliability. In practice, we repeat this procedure three times without flagged data replacement to ensure sufficiency.

**Lexical associations.** Next, we devise a lightweight mutual-information score to capture lexical shortcuts. We estimate surface correlations between simple features $Z$, such as context–option overlap and option length, and the correctness labels $Y$. Formally, for each feature $Z$, we compute its mutual information with $Y$ as $I(Z; Y) = \sum_{z,y} p(z,y) \log \frac{p(z,y)}{p(z)p(y)}$ and normalize by the entropy of $Y$ so that the score lies between $0$ (no association) and $1$ (perfect association). We then average the normalized scores across features to obtain a single measure, $S_{\text{lex}}$. A higher $S_{\text{lex}}$ indicates stronger lexical associations. In practice, we flag a dataset as shortcut-prone if $S_{\text{lex}} \geq 0.15$.

Figure 2: Comparison of GPT-4o zero-shot vs. LLM-discovered shortcuts across 4 ToM benchmarks. The shortcut solution largely outperforms the strong baseline.

### 2.2 KEY FINDINGS AND RECOMMENDATIONS

We audit 8 popular ToM benchmark/datasets spanning both language-only and multimodal domains for shortcuts: ExploreToM (Sclar et al., 2025), OpenToM (?), ToMi (Le et al., 2019), Hi-ToM (He et al., 2023), FANToM (Kim et al., 2023), ToMATO (Shinoda et al., 2025), MMToM (Jin et al., 2024), and MuMA-ToM (Shi et al., 2025). We summarize their properties and show the probing results in Table 1 and Figure 2. In general, 4 of the datasets we probe exhibit shortcuts [1] On these datasets, the shortcut solution scores over 10% higher than the strong GPT-4o baseline without needing any genuine ToM reasoning. We observed that Hi-ToM suffers the most with shortcut solutions, reaching 95% overall and 99% on 3rd and 4th order queries by exploring the simple rule mentioned section 2. With all respect to prior works, we kindly summarize our findings below and hope they serve the future development of robust ToM tuning sets: **1)** For LLM-generated datasets, problems that only require *state tracking* are prone to shortcuts. In contrast, *intention*-related problems, which demand genuine ToM reasoning beyond pure tracking, are much more robust to shortcuts; **2)** Training on shortcut-prone datasets leads to a false sense of ToM and may harm performance. As shown in Figure 1, RFT mixes "Jack's" own observation with the 4th order query being asked, ignoring intermediate ToM; **3)** Practitioners aiming to improve ToM should consider using real-world data or conduct rigorous audits of the synthetic datasets prior to use. In our study, we adopt the four shortcut-free datasets, namely OpenToM (?), ToMATO (Shinoda et al., 2025), MMToM (Jin et al., 2024), and MuMA-ToM (Shi et al., 2025).

---

[1]We note that since these benchmarks are designed primarily for evaluation, this should not be flagged as a major "mistake". However, it would be problematic if they are used for training/tuning as we have shown.

| Method | 3B model | | 7B model | |
|---|---|---|---|---|
| | In-domain | OOD | In-domain | OOD |
| Zero-shot | 49.5 | 38.5 | 62.0 | 43.6 |
| SFT | 96.4 | 32.5 | 95.8 | 34.2 |
| Thinking-RFT | 93.2 | 31.3 | 94.3 | 35.3 |
| No-Thinking-RFT | 95.8 | 32.0 | 96.1 | 34.0 |

Table 2: Results of training on shortcut-prone datasets. In-domain: ExploreToM; OOD: Hi-ToM.

### 2.3 How do shortcuts impact learning?

Figure 1 demonstrates one example of shortcut-induced illogical reasoning. To further clarify the impact of shortcut solutions during training, we conduct a controlled experiment: we train both 3B and 7B models using SFT, Thinking-RFT, and No-Thinking-RFT on ExploreToM (Sclar et al., 2025), and evaluate both in-domain and out-of-domain (OOD) on Hi-ToM (He et al., 2023) following the setup in subsection 4.1. The results are reported in Table 2. The results clearly reveal four major drawbacks of shortcut-prone datasets:

- **Shortcuts invert the ranking of training methods, obscuring their relative strengths.** On ExploreToM, the 7B model exhibits the ordering *No-Thinking-RFT > SFT > Thinking-RFT* (96.1% > 95.8% > 94.3%), which is the exact opposite of our results on shortcut-free datasets (subsection 4.2) where Thinking-RFT is consistently best. This misleading ranking would cause practitioners to prefer the wrong post-training strategy for enhancing ToM.

- **Shortcuts mask the benefits of model scaling.** On shortcut-free datasets, scaling from 3B to 7B yields clear gains (e.g., on OpenToM SFT improves from 77.4% $\rightarrow$ 83.1% and Thinking-RFT from 83.0% $\rightarrow$ 89.1%). In contrast, on ExploreToM both 3B and 7B collapse to similar in-domain accuracy (93–96%), and the 3B model even slightly outperforms 7B under SFT (96.4% vs. 95.8%). We hypothesize that this is because overfitting cheap heuristics do not require much capacity. This also means that when trained on shortcut data, additional capacity brings no benefit and most capacity is wasted.

- **Shortcut-driven training harms generalization and can induce negative transfer.** For the 3B model, post-training on ExploreToM raises in-domain accuracy from 49.5% (zero-shot) to 93–96%, yet Hi-ToM accuracy *drops* from 38.5% to 31–33%. The 7B model shows the same pattern (43.6% zero-shot vs. 34–35.3% after post-training). This is also the opposite of our cross-dataset genearlization results in Table 7 where Thinking-RFT successfully improve performance even cross-dataset. This shows that shortcut data inflates in-domain accuracy and induces a false sense of ToM.

- **Training on shortcut data fails to teach genuine reasoning.** Beyond accuracy, we evaluate reasoning quality with an LLM-as-judge metric (subsection J.2). Chains-of-thought generated by models trained on ExploreToM receive lower scores than zero-shot+CoT, and are far below Thinking-RFT trained on clean data. This confirms that shortcut-prone training chiefly teaches the shortcut itself rather than robust ToM reasoning.

Together, these results suggest that shortcut-prone datasets are unsuitable as ToM post-training sets: they neither cultivate genuine ToM abilities nor yield a faithful comparison of different post-training methods, a cautionary insight we hope will inform future research in ToM and broader community.

## 3 Reinforcement Learning for Robust ToM Post-Training

In this section, we introduce the details about the proposed RFT for ToM below.

**Optimization algorithm.** We mainly follow Deepseek-R1 (Shao et al., 2024; Guo et al., 2025) and adopt the rule-based Group Relative Preference Optimization (GRPO) as our core RL optimization algorithm. Unlike Deepseek-R1, we set the KL coefficient $\beta$ to 0 (Yu et al., 2025) as our experiments indicate that this regularization is unnecessary for ToM reasoning. We also test alternative optimization algorithms such as DAPO (Yu et al., 2025) and GSPO (Zheng et al., 2025) and found them to be slightly better but not worse. We provide more details in the Appendix I.

**Instruction prompts.** We follow Guo et al. (2025) and adopt an instruction prompt that encourages exploration and reasoning before answering: `ToM instruction here. ToM story and question here. Please output your reasoning process in <think>...</think> and the final answer, either a short phrase or an option letter, in <answer>...</answer>`. For SFT, we fix the ToM instructions and ask to output the final answer directly: `Please output your final answer, either a short phrase or an option letter directly.`

**Reward functions.** Following Guo et al. (2025), our default reward function combines a format reward $R_{\text{format}}$ and an accuracy reward $R_{\text{accuracy}}$. The format reward encourages the model to first produce reasoning before giving a verifiable answer: $R_{\text{format}} = 1$ if the output includes both `<think></think>` and `<answer></answer>` tags, and 0 otherwise. The accuracy reward evaluates the extracted answer within the answer tags, assigning $R_{\text{accuracy}} = 1$ if it matches the ground truth, and 0 otherwise. The overall reward is $R_{\text{overall}} = R_{\text{format}} + R_{\text{accuracy}}$. Additionally, we explore a ToM-specific reward $R_{\text{tom}}$ in Appendix H designed to further streamline ToM reasoning.

## 4 EXPERIMENTS

In this section, we introduce our main results and findings on ToM tasks. We show that when trained on shortcut-free data, RFT outperforms SFT considerably in three different ToM contexts tested: narrative, conversational, and multimodal. Moreover, it shows much better generalization in lower- to higher-order reasoning and unseen environments and is robust to counterfactual manipulations. All results are obtained by averaging *three* randomly seeded runs.

### 4.1 EXPERIMENTAL SETTING

**Datasets and benchmarks.** ToM evaluation requires handling diverse scenarios across *narrative vs. conversational* inputs, *language-only vs. vision-language* modalities, and higher-order reasoning (beyond 1st order). We therefore experiment with the 4 *shortcut-free* datasets that cover all scenarios: **ToMATO** (Shinoda et al., 2025) (conversational), **OpenToM** (?) (narrative), and **MMToM** (Jin et al., 2024) / **MuMA-ToM** (Shi et al., 2025) (multimodal). Importantly, these also cover a range of ToM categories, belief, desire, intention, location, multihop, attitude, etc. We report dataset and category-level accuracy for all settings except multimodal. For OpenToM, we additionally report the macro average stratified by ToM order. We provide more in-depth details on the datasets and sampling methods in Appendix E and experiments on mixed-training in the subsection K.2 and note that the performance trend is preserved.

**Models.** We use Qwen2.5-7B-Instruct (Qwen, 2024) and its VL variant (Qwen, 2025) as the default model for all experiments. We compare RFT against SFT and the zero-shot baseline. We also compare with state-of-the-art inference-time algorithms: SimTom (Wilf et al., 2024) and AutoToM (Zhang et al., 2025b), where their details are in Appendix E. Moreover, we demonstrate scaling behavior with the smaller 3B variant.

### 4.2 MAIN RESULTS

**Result 1: ToM benefits more from RL-based post-training compared to SFT across all contexts.** We find that RFT consistently surpasses the SFT baseline in every evaluation setting. On average, RFT improves over SFT by 6.0% in narrative tasks, 2.08% in conversational tasks, and 10.55% in multimodal tasks. These gains highlight that rule-based RL post-training is effective at eliciting ToM knowledge and skills in foundation models. Importantly, this trend holds not only for the 7B backbone but also for the smaller 3B variant, suggesting that benefits of RL tuning are robust across model scales.

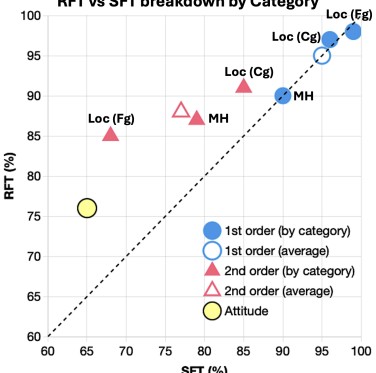

Figure 3: Detailed RFT vs. SFT acc. breakdown by category. Entries *above the diagonal line* denote where RFT performs better: **RFT outperforms SFT, particularly in challenging problems: higher-order, attitude, etc.**

Table 3: RFT outperforms SFT and No-Thinking RFT considerably on **OpenToM (narrative)** (?) across location coarse-grained (Loc (Cg)) and location fine-grained (Loc (Fg)), multi-hop (MH), and attitude (Att.). "First" and "second" denote first and second-order ToM queries. We also report the overall average and that of the $1^{st}$ and $2^{st}$ queries. The best results are highlighted .

| Method | Loc (Cg) | | Loc (Fg) | | MH | | Att. | Avg($1^{st}$/$2^{st}$/overall$_{\Delta \text{ vs. SFT}}$) |
|---|---|---|---|---|---|---|---|---|
| | First | Second | First | Second | First | Second | | |
| **Qwen-2.5-3B Models** | | | | | | | | |
| Zero-shot | 51.00 | 50.00 | 28.00 | 15.00 | 53.00 | 48.00 | 39.00 | 44.00 / 37.67 / 40.57$_{\downarrow 39.90}$ |
| SFT | 100.00 | 81.00 | 92.00 | 56.00 | 88.00 | 76.00 | 49.00 | 93.33 / 71.00 / 77.43 |
| No-Thinking RFT | 81.00 | 50.00 | 81.00 | 63.00 | 56.00 | 55.00 | 35.00 | 72.67 / 56.00 / 60.14$_{\downarrow 17.29}$ |
| Thinking RFT | 99.00 | 88.00 | 94.00 | 67.00 | 91.00 | 85.00 | 57.00 | 94.67 / 80.00 / 83.00$_{\uparrow 5.57}$ |
| **Qwen-2.5-7B Models** | | | | | | | | |
| Zero-shot | 49.00 | 51.00 | 48.00 | 33.00 | 55.00 | 53.00 | 36.00 | 50.67 / 45.67 / 46.43$_{\downarrow 36.71}$ |
| **SimToM (ACL'24)** | 50.00 | 53.00 | 44.00 | 44.00 | 70.00 | 40.00 | 47.00 | 54.67 / 45.67 / 49.71$_{\downarrow 33.43}$ |
| **AutoToM (NeurIPS'25)** | 57.00 | 64.00 | 60.00 | 48.00 | 66.00 | 60.00 | 43.00 | 61.00 / 57.33 / 56.86$_{\downarrow 26.28}$ |
| SFT | 99.00 | 85.00 | 96.00 | 68.00 | 90.00 | 79.00 | 65.00 | 95.00 / 77.33 / 83.14 |
| No-Thinking RFT | 97.00 | 84.00 | 93.00 | 71.00 | 88.00 | 82.00 | 36.00 | 92.67 / 79.00 / 78.71$_{\downarrow 4.43}$ |
| Thinking RFT | 98.00 | 91.00 | 97.00 | 85.00 | 90.00 | 87.00 | 76.00 | 95.00 / 87.67 / 89.14$_{\uparrow 6.00}$ |

Table 4: RFT improves on **multimodal** benchmarks **MMToM** (Jin et al., 2024) and **MuMA-ToM-ToM** (Shi et al., 2025) substantially compared to SFT. For MMToM, we also report the language-only (Lan) results to isolate the effect of vision. Best results in each column are highlighted .

| Method | Train Modality | MMToM | MuMA-ToM | Avg$_{\Delta \text{ vs. SFT}}$ |
|---|---|---|---|---|
| Zero-shot | Lan. | 39.40 | – | – |
| Zero-shot | Lan.+Vis. | 45.00 | 43.30 | 44.15$_{\downarrow 30.60}$ |
| **SimToM (ACL'24)** | Lan.+Vis. | 50.60 | 49.60 | 50.10$_{\downarrow 24.65}$ |
| **AutoToM (NeurIPS'25)** | Lan.+Vis. | 56.90 | 59.10 | 58.00$_{\downarrow 16.75}$ |
| SFT | Lan. | 73.02 | – | – |
| SFT | Lan.+Vis. | 74.30 | 75.20 | 74.75 |
| Thinking-RFT | Lan. | 78.50 | – | – |
| Thinking-RFT | Lan.+Vis. | 83.30 | 81.10 | 82.20$_{\uparrow 7.45}$ |

**Result 2: RFT excels particularly in complex scenarios such as higher-order reasoning and multimodal input.** As shown in Figure 3, RFT's advantage over SFT comes almost entirely from second-order questions: while first-order accuracy shows no gain ($+0.0\%$), RFT outperforms SFT by $+10.3\%$ on second-order, highlighting its strength in recursive belief attribution (e.g., "what does A think that B believes?"). This suggests that RFT guides models toward genuine reasoning trajectories rather than shallow heuristics; this is further supported by analysis of its attention pattern (subsection 5.3). Gains are also amplified in multimodal settings, where RFT surpasses SFT by $+10.55\%$, compared to $+6.0\%$ in narrative and $+2.08\%$ in conversational contexts, showing that reinforcement-driven exploration not only enhances recursive reasoning but also grounds it more effectively in perceptual evidence.

**Result 3: RFT enables larger gains on mind-state related questions compared to belief/knowledge.** We observe that although RFT improves over SFT on most categories across both benchmarks, the relative gains differ between tracking-oriented and mind-oriented tasks. On TOMATO (Qwen-2.5-VL-7B), RFT yields larger gains on *intention* (+3.6 pts) and *desire* (+3.3 pts), while the lift is more modest on *belief* (+1.3) and *knowledge* (+2.8), with the only exception being *emotion*, where SFT slightly outperforms RFT (–1.2 pts). On OPENTOM, the clearest improvement is on *attitude* (mind-state), with +11 pts for 7B and +8 pts for 3B, while tracking-style categories such as *location* and *multi-hop* see smaller but steady improvements. Overall, these results suggest that RFT is effective at strengthening inference over latent *mental* states (preferences, intentions).

## 4.3 GENERALIZATION

AI agents in the real world often interact with multiple characters and encounter unexpected events or environments, which demand good generalization.

**Generalization from lower- to higher-order.** We evaluate the *depth* of ToM reasoning, i.e., whether models trained only on first-order questions can solve second-order ones that require recursive inference over nested beliefs (e.g., whether $A$ believes that $B$ thinks favorably about an

Table 5: Generalization from 1st- to 2nd-order ToM on **OpenToM** and **ToMATO**. Best results are **bolded**. RFT shows 9% and 3.04% higher accuracy on unseen second-order queries on OpenToM and ToMATO. We provide fully detailed category-level scores in subsection K.2.

| Method | First Order (*Seen*) | | ↪ Second Order (*Unseen*) | |
|--------|---------|---------|---------|---------|
| | OpenToM | ToMATO | OpenToM | ToMATO |
| Zero-shot | 50.67 | 72.96 | 45.67 | 62.22 |
| SFT | 93.00 | 88.08 | $65.33_{\downarrow 27.67}$ | $81.74_{\downarrow 6.34}$ |
| RFT | **95.00** | **89.32** | $\mathbf{74.33}_{\downarrow 20.67}$ | $\mathbf{84.78}_{\downarrow 4.54}$ |

Table 6: Generalization to unseen environments on **MMToM**. Results are reported on six environments with overall average. Overall, RFT shows better retention with a smaller average drop.

| Method | Seen | ↪Unseen | | | | | Avg |
|--------|------|---------|---------|---------|---------|---------|-----|
| | Apartment | Andersen Tales | Ancient Egyptian | Outer Space | Wild West | Medieval Castle | |
| Zero-shot | 45.00 | 48.00 | 47.00 | 52.00 | 46.00 | 46.00 | 47.80 |
| SFT | 74.30 | $67.10_{\downarrow 7.20}$ | $68.92_{\downarrow 5.38}$ | $67.60_{\downarrow 6.70}$ | $66.21_{\downarrow 8.09}$ | $65.12_{\downarrow 9.18}$ | $68.99_{\downarrow 7.31}$ |
| RFT | **83.30** | $\mathbf{81.24}_{\downarrow 2.06}$ | $\mathbf{79.80}_{\downarrow 3.50}$ | $\mathbf{84.40}_{\uparrow 1.10}$ | $\mathbf{78.92}_{\downarrow 4.38}$ | $\mathbf{79.93}_{\downarrow 3.37}$ | $\mathbf{80.86}_{\downarrow 2.44}$ |

event $e$). This setting tests whether models can track multiple agents' perspectives without confusion. We implement it by training on first-order questions and evaluating on second-order questions in OPENTOM and TOMATO.

**Generalization to unseen environments.** We evaluate the *breadth* of ToM reasoning under distribution shift in context. Advanced ToM must reason about action sequences beyond surface heuristics: opening a fridge may imply hunger in a household but medicine storage in a hospital, leading to different mental states. To probe this, we extend OPENTOM with three novel contexts—*Hospital*, *Museum*, and *Airport/Airplane Cabin*—via GPT-5 rewrites that preserve the underlying ToM logic (details in subsection K.2). We further adopt five procedurally generated environments from MM-TOM (Jin et al., 2024): *Ancient Egypt*, *Outer Space*, etc., with *Apartment* as the seen baseline.

**Cross-dataset generalization.** We further examine whether post-training transfers to other ToM benchmarks. Concretely, we train on OPENTOM and evaluate on TOMATO and EXPLORETOM. We note that since these datasets differ in narratives, question templates, and labeling schemes, it is not a direct rendition of the transferability of ToM skill alone.

**Result 4: RFT shows much better generalization in lower- to -higher-order reasoning, unseen environment, and distinct datasets compared to the SFT baseline.** As shown in Table 5, when trained exclusively on first-order ToM questions, RFT reaches 74% on second-order questions, outperforming zero-shot and SFT baselines by +28.67% and +9% on average, respectively, when their in-domain (first-order) performance sees little difference (95% vs. 93%). On the harder location(fg) category, where the explicit end location of the object is asked, this

Table 7: Cross-dataset generalization from OpenToM to ToMATO and ExploreToM. Thinking-RFT improves accuracy on both datasets, whereas SFT yields little or even negative gain compared to zero-shot.

| Method | ToMATO | ExploreToM |
|--------|--------|-----------|
| Zero-shot | 67.5 | 62.0 |
| SFT | 56.8 | 63.5 |
| Thinking-RFT | 70.4 | 71.0 |

gap widens to 15% compared to SFT (Table 14). We note that there is a 20% drop from first-order due to the nature of the increased difficulty. Nevertheless, these results together demonstrate the strong generalization of RFT in ToM.

Similar strength in generalization is also observed in unseen domain settings, as reported in Table 6. In OpenToM, RFT achieves 88.22% (vs. 75.99% for SFT) on average across the three unseen domains, a marginal 0.92% drop from in-domain, compared to the 7.15% drop for SFT. On MMToM, the same pattern is observed: RFT sees a 2.4% degradation for the 5 novel contexts compared to in-domain, while SFT shows a 7.11% decrease. To summarize, RFT develops ToM that is robust to environment shift, unlocking more real-world potentials. We note that better generalization of RFT is consistent with prior work (Lai et al., 2025; Li et al., 2025), but they are done in different settings.

Moreover, across cross-dataset experiments reported in Table 7, Thinking-RFT is the only method that consistently improves performance on both OOD datasets when trained on OpenToM: accuracy

Table 8: **Robustness to counterfactual modifications.** Accuracy (%) on OPENTOM and TOMATO before and after counterfactual rewrites. We also report a **CFC-LB↑**, the conditional flip consistency lower bound.

| Method | OpenToM | | | ToMATO | | |
|---|---|---|---|---|---|---|
| | Original | Counterfactual | CFC-LB↑ | Original | Counterfactual | CFC-LB↑ |
| Zero-shot | 45.00 | 46.00 | **0.00** | 69.00 | 68.00 | **53.62** |
| SFT | 83.14 | $66.00_{\downarrow 17.14}$ | **59.11** | 87.92 | $73.80_{\downarrow 14.12}$ | **70.20** |
| RFT | **89.14** | $\mathbf{81.24_{\downarrow 7.90}}$ | **78.95** | **90.00** | $85.33_{\downarrow 4.67}$ | **83.70** |

Table 9: Quantitative reasoning quality evaluation on **OpenToM** and **ToMATO**. "LLM-Judge" reports LLM-based ratings of the reasoning traces along three axes: LC=logical consistency, F=faithfulness, and E=efficiency. Maximum score for each category is 10.

| Method | OpenToM | | ToMATO | |
|---|---|---|---|---|
| | Acc | LLM-Judge (LC/F/E) | Acc | LLM-Judge (LC/F/E) |
| Zero-shot | 46.4 | 4.3 / 2.2 / 8.0 | 67.5 | 5.6 / 4.2 / 7.6 |
| Thinking-RFT | 89.1 | 9.1 / 9.9 / 6.5 | 90.0 | 9.2 / 10.0 / 7.0 |
| Zero-shot + RFT Reasoning Trace | 74.7 | – | 82.0 | – |

rises from 67.5 to 70.4 on ToMATO and from 62.0 to 71.0 on Explore-ToM, while SFT produces mixed changes. This indicates that the ToM skills acquired via Thinking-RFT on a single shortcut-free dataset remain effective under substantially different benchmarks, whereas SFT offers limited and inconsistent transfer.

### 4.4 ROBUSTNESS TO COUNTERFACTUAL INFORMATION

As discussed in Sec. 4.3, RFT enables transfer of ToM reasoning to new contexts, but it remains unclear whether the learned reasoning relies on true *causal* factors or brittle cues undiscovered in subsection 2.1. To test this, we design a simple robustness experiment based on *counterfactual rewrites*: minimally altering a story (e.g., changing "like" to "dislike") so that the ground truth flips while coherence is preserved. We leave the implementation details to section E.

**Result 5: RFT enables causal factors-anchored reasoning.** Table 8 shows that SFT suffers large accuracy drops (–17.14% and –14.12%), while RFT degrades only about half as much (–7.90%, –4.67%). Moreover, CFC-LB exposes the underlying causal sensitivity: SFT achieves only 0.59/0.70, while RFT reaches 0.79/0.84, closer to the ideal 1.0. We conduct further manual inspection and confirm that most remaining RFT errors stem from edits that introduce residual label ambiguity. Removing these pairs shrinks RFT's degradation, suggesting that RL with verifiable rewards grounds reasoning in causal factors rather than superficial correlations.

## 5 IN-DEPTH ANALYSIS AND DISCUSSION

In Sec. 4.2, we have shown superior performance of RFT, yet it remains unclear *why it helps this much?* We attempt to bridge this gap with in-depth analyses from *three* unique perspectives: **1)** Quantitative reasoning trace and error analysis. **2)** We show that explicit *reasoning-based* tuning is key to ToM post-training by comparing No-Thinking-RFT vs. Thinking-RFT. **3)** We then show that RFT results in a better reasoning strategy by focusing on key information that forms exquisite reasoning patterns by visualizing the attention, explaining why it helps.

### 5.1 QUANTITATIVE ANALYSIS OVER REASONING TRACES.

We employ a three-part analysis to quantify the quality of reasoning traces produced by the RL-trained model: (1) a "Reasoning Trace + Zero-shot" setting, (2) LLM-based judging of trace quality, and (3) manual error analysis. The results are shown in Table 9. Across datasets, feeding only the Thinking-RFT thinking traces (with the final answer removed) to the frozen base model increases accuracy substantially compared to the 0-shot baseline (+28.3% and 14.5% respectively).

This demonstrates that the Thinking-RFT-produced traces are not superficial explanations but contain task-relevant ToM logic and awareness. The LLM-judge scores show the same trend: compared to 0-shot, Thinking-RFT yields substantially higher logical consistency and faithfulness, with only a mild trade-off in efficiency. Finally, our manual inspection reveals that most genuine mistakes (∼50%) arise from mis-tracking the target agent or object (e.g., collapsing second-order queries into first-order reasoning). Notably, ∼15% of errors come from selecting an incorrect final option even when reasoning is logically correct. We provide more details about the setting and representative examples in subsection J.1.

## 5.2 ROLE OF REASONING AND RL IN TOM: THINKING-RFT VS. NO-THINKING-RFT

A key motivation for applying Thinking-RFT in ToM post-training is the assumption that ToM tasks require reasoning to solve, thus, by encouraging unrestrained thinking before answering, Thinking-RFT provides a way to incentivize ToM in LLMs from knowledge gained in pre-training for free. To further test this assumption, we propose a simple *No-Thinking-RFT* experiment by isolating the effect of reasoning. Specifically, No-Thinking-RFT differs from Thinking-RFT in that 1) we ask in the prompt `<same ToM instructions. Please output your final answer, either a short phrase or an option letter, directly without anything else`, and 2) we only use the accuracy reward $R_{accuracy}$ without $R_{format}$. Indeed, previous works have shown that for tasks such as multimodal perception or medical QA, explicit thinking for Thinking-RFT is unnecessary at best, harmful at worst (Seed et al., 2025; Li et al., 2025).

As shown in Table 3 and Table 15, our results show that although No-Thinking RFT still improves, it lags the reasoning variant substantially by 10.43% and 3.08% on OpenToM (**?**) and ToMATO (Shinoda et al., 2025), respectively. This is also evident from Figure 4. We hypothesize that this is due to the logical turns in ToM questions that are hard to master in the latent space alone. These results indicate that for ToM, RFT must be combined with reasoning to be effective. More importantly, it suggests that **theory of mind benefits explicitly from reasoning-based finetuning**.

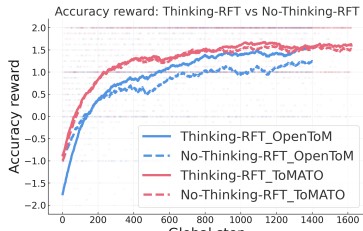

Figure 4: Accuracy reward for Thinking-RFT vs. No-Thinking-RFT on ToMATO and OpenToM.

## 5.3 HOW DOES RFT AND THINKING HELP?

To probe why RFT with explicit reasoning helps, we visualize last-layer (27) attention maps under four settings: Thinking-RFT, Zero-shot+CoT, No-Thinking-RFT, and SFT. For each model, Figure 5 plots a heat map whose *y-axis* index the generated output tokens in the reasoning/answer, and whose *x-axis* index the input sentences (we aggregate tokens within a sentence and label each column by its first word). Thus, each row shows which sentence the model focuses on when producing a particular token; lighter colors indicate higher average attention over all heads.

The example story of Figure 5 is about Matthew and Grant, and the task is to classify Matthew's attitude (positive / neutral / negative). Although the narrative contains several distractor actions, the correct label depends on two "causal hinge" sentences: (i) Matthew loves bananas and wants easy access to them, and (ii) the banana is moved away from him. In the Thinking-RFT panel, attention forms clear vertical bands over *exactly these two sentences* when the model generates the crucial parts of its chain-of-thought and final answer, indicating that it repeatedly returns to the *minimal causal cues* needed for the decision. In contrast, the Zero-shot+CoT panel displays much more *diffuse attention* that often drifts to *irrelevant details* about Grant's behavior. This qualitative pattern supports our quantitative findings: after RFT on shortcut-free data, the model's reasoning becomes better aligned with the true causal structure of the story, which is precisely the type of grounding required for robust ToM reasoning. We further plot the attention map of No-Thinking-RFT and SFT in Figure 10; similarly, both models fail to capture the anchor sentences and instead over-focus on the latter irrelevant parts of the input. To substantiate these observations, we further conduct a small-scale quantitative study in subsection J.3. Together, these analysis suggests **Thinking-RFT effectively incentivizes reasoning that prioritizes crucial ToM-specific information that is robust to irrelevant information**.

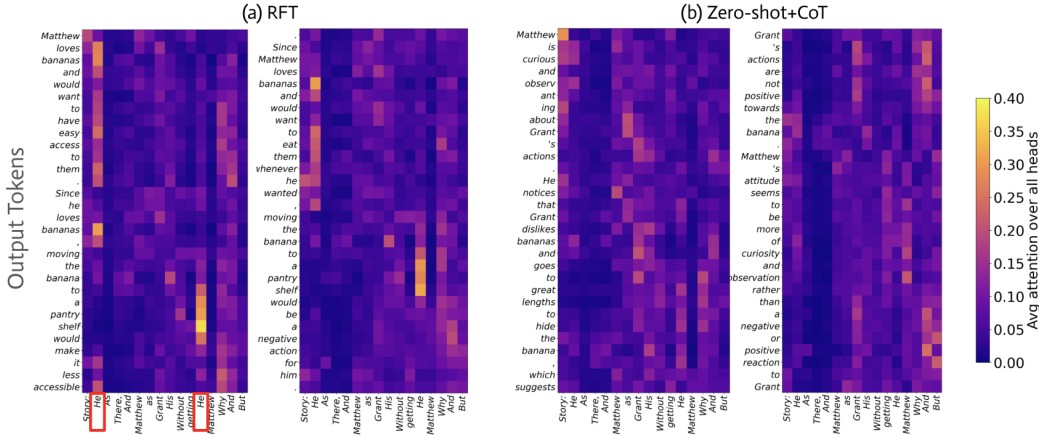

Figure 5: Attention visualization of RFT vs. Zero-shot+CoT at the last layer, averaged across all heads. The x-axis and y-axis denote input and output tokens, respectively, with attention weights row-wise normalized for relative comparison. **RFT shows stronger focus on key causal information linking characters' intentions to narrative events**, indicating more effective ToM thinking.

## 6 RELATED WORK

**Theory of Mind in foundation models.** ToM evaluation has expanded significantly in foundation models. Early efforts include short, bias-controlled QA (TOMI) (Le et al., 2019) and social commonsense tasks (e.g., SocialIQa) (Sap et al., 2019), later extended to larger batteries with controlled templates (BIGTOM) (Gandhi et al., 2023), interaction asymmetry (FANTOM) (Kim et al., 2023), long-form narratives and psychological states (OPENTOM) (**?**), and fine-grained coverage (TOMBENCH) (Chen et al., 2024). ExploreToM and Hi-ToM further test story-style reasoning, while multimodal datasets such as MMToM-QA and MuMA-ToM (Jin et al., 2024; Shi et al., 2025) extend ToM assessment beyond text. More recently, ToMATO (Shinoda et al., 2025) targets intention-based ToM. Overall, these works focus on benchmarking rather than training or post-training methods for improving ToM.

**Rule-based reinforcement fine-tuning for post-training.** Rule-based RFT has proven promising for LLMs (Guo et al., 2025; Jaech et al., 2024; Kimi et al., 2025), surpassing SFT. Extensions to MLLMs (Liu et al., 2025; Shen et al., 2025; Zhou et al., 2025; Huang et al., 2025; Meng et al., 2025; Chen et al., 2025) aim to replicate Deepseek-R1 phenomena such as longer responses and emergent "aha" moments. Closest to our work is Lu et al. (Lu et al., 2025), who also study ToM post-training with RL and SFT but arrive at different conclusions: on *shortcut-prone* benchmarks (Hi-ToM, Explore-ToM) they report that both RL and SFT quickly exceed 90% accuracy and SFT outperforms RFT. This matches our own observations in our *shortcut experiment* (subsection 2.3). In contrast, our work first audits ToM datasets to identify and remove shortcut-heavy ones, and then shows that on cleaned ToM suite Thinking-RFT consistently outperforms SFT and No-Thinking-RFT in accuracy and generalization. Together, these results suggest that the discrepancy is largely explained by *differences in benchmark design and dataset selection*, and highlight the importance of careful data filtering and shortcut auditing when conducting reasoning-focused post-training.

## 7 CONCLUSION

In this work, we asked whether rule-based reinforcement fine-tuning can reliably elicit theory-of-mind (ToM) abilities in foundation models, and found that it can—provided the data does not admit cheap shortcuts. We first showed that several popular ToM benchmarks are shortcut-prone and proposed a simple auditing framework to identify and avoid such cases. On a cleaned suite of narrative, conversational, and multimodal ToM datasets, Thinking-RFT consistently outperforms SFT and No-Thinking-RFT in accuracy, higher-order and cross-dataset generalization, and counterfactual robustness. Analyses of reasoning traces and attention patterns further reveal that Thinking-RFT-trained models ground their chains of thought on key causal anchors rather than superficial cues.

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

# APPENDIX

In the supplementary material, we provide additional algorithmic details, reward design, extended results, hyperparameter settings, and discussions on ethical, reproducibility, and LLM usage considerations.

This supplementary is organized as follows:

- **Appendix A**: Limitations
- **Appendix B**: Ethics Statement
- **Appendix C**: Reproducibility Statement
- **Appendix D**: Use of Large Language Models
- **Appendix E**: Implementation Details.
- **Appendix F**: Group Relative Policy Optimization (GRPO)
- **Appendix H**: ToM-specific Reward Design
- **Appendix G**: Shortcut
- **Appendix J**: Additional Analysis
- **Appendix I**: Additional RL Optimization Algorithms
- **subsection K.2**: Additional Experimental Results

## A LIMITATIONS

We acknowledge several limitations: 1) our reinforcement learning experiment does not include a theory of mind specific cold-start. Although the Qwen2.5-7B-Instruct model has already been tuned to follow instructions, they have likely not been tuned on CoT-enabled ToM dataset, hence the effect of cold-start + RFT remains to be explored. 2) Our evaluation is still done in the synthetic setting by evaluating model performance on QA pairs. Recent work (Riemer et al., 2025) has pointed out that ToM may need to be evaluated as a foundation model's functional capability (i.e., their ability to act and execute tasks with ToM in mind). 3) Although our framework catches most of the shortcuts, there might still be undiscovered ones. 4) The highest order of ToM query in our study is second. While this is more complex than first order, it is still far from the order to which agents may encounter in the real world. We did not experiment on higher-than-2 order data as there is none available (all available ones contain heavy shortcuts). To this end, we look forward to seeing more high quality dataset get released, especially on shortcut-free higher order queries.

## B ETHICS STATEMENT

This work focuses on advancing theory of mind (ToM) evaluation and training for large language models. We only use publicly available datasets (ExploreToM, HiToM, ToMi, MMToM, etc.) that have been released with appropriate research licenses, and no personally identifiable information (PII) or sensitive human subject data is involved. Our models are evaluated on reasoning benchmarks rather than deployed in real-world applications, thus posing minimal risk of harm. We acknowledge that ToM research may potentially be misused for manipulative purposes (e.g., persuasive systems or deceptive agents), and we encourage responsible use of this research strictly within scientific and educational contexts.

## C REPRODUCIBILITY STATEMENT

We are committed to reproducible science and reproducible research; all code and model weights will be publicly released.

## D    USE OF LARGE LANGUAGE MODELS

Large language models (LLMs) were used in this work only as general-purpose assistive tools. Specifically, LLMs (e.g., ChatGPT) were employed to support grammar polishing, figure caption refinement, and formatting suggestions during manuscript preparation. They were not used for generating novel research ideas, running experiments, or producing results. All scientific contributions, including dataset design, model implementation, and analysis, were conceived and conducted by the authors. Therefore, while LLMs facilitated writing clarity, they are not considered contributors to the intellectual content of this paper.

## E    IMPLEMENTATION DETAILS

**Details on the LLM/Agent-guided rules for identifying shortcuts in datasets.**    In our shortcut-auditing framework, the "stratified seed set" $D_{\text{seed}}$ is a small labeled subset of each dataset consisting of full $(x, y)$ QA pairs, where $x$ includes the story, question, and options, and $y$ is the correct label. We construct $D_{\text{seed}}$ by sampling examples roughly uniformly across (i) ToM category (e.g., belief / location / attitude), (ii) ToM order (1st/2nd/3rd/4th where available), and (iii) answer label, so that the LLM sees a representative mix of problem types rather than a skewed slice.

Given $D_{\text{seed}}$, we prompt a frozen LLM to enumerate candidate shortcut hypotheses (e.g., "always predict the object's location when the outer-most agent leaves", "choose the option whose text overlaps most with the last mentioned container"). Each hypothesis is then implemented as a deterministic "zero-update" heuristic

$$h_k : x \mapsto \hat{y}$$

(i.e., a small piece of rule-based code with no learning) and evaluated on a disjoint held-out probe set $D_{\text{probe}}$. We compute its accuracy $A(h_k)$, compare it against a random/majority baseline $A_0$, and accept $h_k$ as a plausible shortcut if

$$A(h_k) - A_0 \geq \delta_{\text{abs}}$$

(with $\delta_{\text{abs}} = 0.2$ by default) and the improvement is statistically significant ($p < 0.05$ under a binomial test).

We repeat this procedure multiple times and flag a dataset as "causal-shortcut-prone" if any such heuristic meets the threshold $\delta_{\text{abs}} = 0.2$.

**Datasets and benchmarks.**    Real-world agents need to possess ToM intelligence not only in reading off event narrations but also in real conversations and even in multimodal environment that potentially involves multiple agents. They need to correctly track the agent's mental beliefs and infer their mental states. Thus it is imperative to consider different input scenarios (*narrative vs. conversational*), modalities (*language-only vs. vision-language*), ToM orders (*higher order, beyond 1st order*), ToM categories (*mental belief vs. mental state*), and belief status (*false beliefs*). With these considerations, we conduct experiments on three datasets that represent different ToM contexts:

- **ToMATO (Shinoda et al., 2025).** A **conversational** style dataset where ToM is assessed by answering multiple-choice questions based on given conversations among the agents. These questions span 5 categories: belief, desire, emotion, intention, and knowledge, each up to order 2 (second-order ToM). We generate 5k problems and uniformly select problems from each category, which results in a train set of 2.5k. We evaluate on the original benchmark and report category-level accuracy and a macro average.

- **OpenToM (?).** A **narrative** style dataset where ToM is assessed by answering multiple-choice or free form questions based on the given story of agents' interactions. These questions span 4 categories, coarse-grained location, fine-grained locations, multihop, and attitude, each up to order 2 (second-order ToM). OpenToM contains over 20k problems, of which we sample 2.8k problems uniformly from each category to form the train set. We then uniformly sample 100 questions from each category to form an evaluation set. We report category-level accuracy within each of the two ToM orders.

- **MMToM (Jin et al., 2024) and MuMA-ToM(Shi et al., 2025)**. **Multimodal** style datasets with vision and language input. ToM is assessed by watching a relative video along with fullpartial discriptions and answer ToM based questions. We use a default sampling rate of

16 FPS. We randomly split each dataset by 80% for training and 20% for evaluation and report dataset-level accuracy.

**Models.** We use Qwen2.5-7B-Instruct (Qwen, 2024) and Qwen2.5-VL-7B-Instruct (Qwen, 2025) as the default model for all LLMs and multimodal-based experiments, respectively. We also use Qwen2.5-3B-Instruct to demonstrate scaling behavior in smaller models. We note that the "instruct" version of the models is selected as they are already trained to follow instructions, thus can directly learn the task without spending time on learning instruction following. We empirically found that base models Qwen2.5-3B/7B can reach the same accuracy just with longer training.

**Training.** For RFT, we use a batch size of 1 per GPU with a rollout size of 8 and with gradient checkpointing. We set $\beta$ to 0 to omit KL divergence regularization in for tuning LLMs and 0.04 for VLMs. We use a starting learning rate of 2e-5 and train for 2 epochs. For SFT, we use a batch size of 2 and a starting learning rate of 1e-6 and train for 2 epochs. All experiments are conducted on $4\times$ NVIDIA H100 GPUs. For all experiments, we run the same setting without hyperparameter change 3 times and report the average.

**Other baselines.** Besides Zero-shot and SFT, we also select popular inference-time algorithms SimToM Wilf et al. (2024) and AutoToM Zhang et al. (2025b) for comparison:

- **SimToM** keeps the backbone LLM frozen and applies a two-stage prompting pipeline that first rewrites the narrative purely from the queried agent's perspective (filtering out events the agent does not observe or is not told about) and then answers the ToM question based solely on this perspective-specific story, thereby improving belief tracking without any additional training.

- **AutoToM** treats ToM as a model-based reasoning problem and uses the LLM in a multi-step Bayesian scaffold that parses the story into structured events, proposes candidate mental-state hypotheses (e.g., beliefs, desires), scores their consistency with the agent's observations and actions, and finally selects the highest-scoring hypothesis to produce the answer, again operating entirely at inference time around a strong frozen backbone.

**Robustness to counterfactual rewrites.** As discussed in Sec. 4.3, Thinking-RFT enables transfer of ToM reasoning to new contexts, but it remains unclear whether the learned reasoning relies on true *causal* factors or brittle cues undiscovered in subsection 2.1. To test this, we design a simple robustness experiment based on *counterfactual rewrites*: minimally altering a story (e.g., changing "like" to "dislike") so that the ground truth flips while coherence is preserved. Concretely, on OPENTOM and TOMATO, we used GPT-5 to identify causal tokens, rewrite them and validate, and evaluate the best performing models on the newly formed paired examples. Besides accuracy, we also report CFC-LB (lower bound on conditional flip consistency), a traditional robustness metric, given by $\text{CFC-LB} = \frac{\text{Acc}_{\text{orig}} + \text{Acc}_{\text{cf}} - 1}{\text{Acc}_{\text{orig}}}$. This metric reflects how reliably models flip their predictions when the causal hinge changes; a higher value correlates with better sensitivity.

**Choice of model.** The choice of "instruct" models is primarily to save budget on learning instruction following, as we empirically found that base models can reach the same accuracy with longer training. Training/hyperparameter details are in Appendix E.

**Choice of SFT Baseline.** In all our experiments, the SFT baseline is trained with *label-only* supervision: the model receives the story, question, and options as input and is trained to predict the correct answer label, without any chain-of-thought annotations. This design choice keeps the supervision budget comparable to Thinking-RFT, which also relies only on verifiable final answers and rule-based rewards rather than additional human-written or teacher-generated reasoning traces. It therefore reflects a realistic setting where collecting high-quality ToM reasoning traces (either manually or via a stronger teacher model) is costly and often impractical. We note that one could, in principle, define a stronger "CoT-SFT" baseline trained on explicit reasoning traces. However, this would require a substantially richer annotation pipeline (and typically a more capable teacher model), making the comparison with Thinking-RFT less fair from a supervision-cost perspective

and less representative of typical deployment scenarios. Our focus here is to isolate the effect of RL with verifiable rewards under a matched supervision budget; studying CoT-augmented SFT is an interesting direction for future work but orthogonal to our main claims.

# F  GROUP RELATIVE POLICY OPTIMIZATION (GRPO)

**Overview.** Reinforcement learning (RL) based algorithms such as PPO (Schulman et al., 2017) and GRPO (Shao et al., 2024) have recently emerged as powerful fine-tuning and alignment strategies for enhancing reasoning capabilities of large models. Unlike supervised fine-tuning (SFT), which directly maximizes likelihood over labeled data, RL-based methods optimize the policy gradient with respect to reward signals. This allows models to explore a larger solution space and to adapt their behaviors according to external alignment objectives. GRPO can be seen as a lightweight variant of PPO: it retains the policy-gradient and clipping framework but introduces group-wise estimation and rule-based rewards, making it more resource-efficient while still effective in eliciting structured thinking behaviors.

**Definition.** Formally, let $P(Q)$ denote the training question set, and let $q \in P(Q)$ be a sampled question at iteration $t$. For each question, the old policy $\pi_{\theta_{\text{old}}}$ generates a group of $G$ candidate responses $\{o_i\}_{i=1}^{G}$. The new policy $\pi_{\theta_{\text{new}}}$ is updated relative to $\pi_{\theta_{\text{old}}}$, while a frozen reference model $\pi_{\theta_{\text{ref}}}$ (typically the base MLLM) anchors the update. The GRPO objective is defined as:

$$
\mathcal{J}_{\text{GRPO}}(\theta) = \mathbb{E}_{q \sim P(Q),\{o_i\}_{i=1}^{G} \sim \pi_{\theta_{\text{old}}}} \Bigg[ \frac{1}{G} \sum_{i=1}^{G} \Bigg(
$$
$$
\min\Bigg( \frac{\pi_{\theta_{\text{new}}}(o_i \mid q)}{\pi_{\theta_{\text{old}}}(o_i \mid q)} A_i, \text{clip}\Big( \frac{\pi_{\theta_{\text{new}}}(o_i \mid q)}{\pi_{\theta_{\text{old}}}(o_i \mid q)}, 1 - \epsilon, 1 + \epsilon \Big) A_i \Bigg)
$$
$$
- \beta\, \mathbb{D}_{\text{KL}}\big( \pi_{\theta_{\text{new}}} \,\|\, \pi_{\theta_{\text{ref}}} \big) \Bigg] \tag{1}
$$

where $\frac{\pi_{\theta_{\text{new}}}(o_i|q)}{\pi_{\theta_{\text{old}}}(o_i|q)}$ is the policy ratio, $A_i$ is the estimated advantage, and $\epsilon$ is the clipping threshold. The KL penalty (Kullback & Leibler, 1951) ensures that the updated policy does not drift too far from the reference model.

**Key differences from PPO.** PPO relies on a critic network to estimate the value function and compute the advantage of a sampled response. In contrast, GRPO removes the critic and instead computes a *relative* advantage within a group of responses. Specifically, a batch of $G$ responses is sampled for the same question, their rule-based rewards are normalized, and the relative scores are used as group-level advantages (Shao et al., 2024; Guo et al., 2025). Furthermore, instead of training a reward model (as in RLHF), GRPO employs a fixed set of rules as the reward function, which substantially reduces both annotation cost and computational burden. As a result, GRPO has been reported to achieve up to 50% higher efficiency compared to PPO while still preserving alignment performance (Shao et al., 2024).

**Intuition.** The key intuition behind GRPO is that relative comparison within a group is often easier and more stable than absolute reward estimation. By rewarding responses that are better than their peers, GRPO enforces a form of rank-based learning that amplifies useful behaviors while suppressing undesirable ones. Combined with rule-based reward design, this makes GRPO particularly appealing for reasoning-intensive tasks where reliable reward modeling is challenging. In our work, GRPO provides a principled and efficient way to encourage explicit thinking traces while avoiding the overhead of large reward models or value-function critics.

# G  SHORTCUT

In this section, we summarize the key shortcuts of each benchmark. We highlight only the most impactful ones; full probing results can be reproduced following our framework.

**ExploreToM.** A large fraction of the benchmark can be captured by just two straightforward heuristics. The first, *Only-room*, applies to queries of the form "In which room . . . ?" If the story mentions only one room, or strongly emphasizes a particular room, then this is almost always the correct answer. Within its coverage, this rule attains 98% accuracy and applies to 58% of the dataset. For instance, when the narrative mentions only the "green room," all room queries resolve to "green room." The second heuristic, *Last-container*, covers container-tracking questions by predicting the most recently mentioned container, independent of belief states. This rule achieves 72% accuracy over 42% of the data. For example, if a banana is last placed in the "cupboard," the answer is consistently "cupboard." Together these two rules yield 87% overall accuracy; incorporating a simple *At-beginning* rule (using the first container mentioned) increases this to 91%.

**ToMi.** The ToMi dataset is similarly shaped by procedural patterns. Two rules are especially predictive. When the query includes "really" or "now," the answer almost always corresponds to the final container in the sequence, yielding 100% accuracy. For instance, if an apple ends up in the "drawer," the question "Where is the apple really?" is answered "drawer." Conversely, when the question refers to the "beginning," the correct response is the first container mentioned, with 98% accuracy. For example, if the story starts with "The toy is in the box," then "Where was the toy at the beginning?" is answered "box." Remarkably, these two heuristics alone reach 75% aggregate accuracy, without the need to track agents' beliefs.

**FANToM.** In FANToM, the dominant regularities are lexical rather than procedural. For yes/no queries of the form "Does NAME know . . . ?", answering "yes" whenever the queried name appears in the `short_context` achieves 85% accuracy. For two-choice `factQA` items, another rule proves perfectly reliable: if one option contains the phrase "does not provide," the correct answer is always the other option (100% accuracy). For instance, between "The text does not provide information about Bob" and "Bob lives in Paris," the latter is invariably correct. Taken together, these two lexical heuristics deliver 86% accuracy on the examples where they apply.

# H  TOM-SPECIFIC REWARDS

Table 10: Performance of ToM-specific rewards on **OpenToM** and **ToMATO**.

| Method | OpenToM | ToMATO |
|---|---|---|
| ToM-specific Rewards | 90.56 | 91.92 |

Designing a generic ToM reward is, in fact, not as simple. As we discussed in section 2, ToM questions can be broken down into two groups: 1) those that can be solved by using pure tracking, 2) those that can be solved only by reasoning on top of tracking. Hence, developing a ToM reward needs to account for both. However, one simple ToM reward can be derived from the error analysis. In our failure cases, we observe that one prominent type of error is the mismatch between the target and the agent-of-mind. For example, when we ask "What does A think about B knows something", the intended reading is that *A* is the agent whose mental state we query (the *agent-of-mind*), and *B* is the entity about which that mental state is ascribed (the *target*). In higher-order reasoning, this mapping recurses (e.g., "A thinks that B thinks that C knows . . . "), and small confusions about which symbol occupies the agent-of-mind vs. target slot at each level can silently derail an otherwise coherent reasoning trace. Motivated by this, we refine the default reward to explicitly check the structural roles before scoring the answer.

Following Guo et al. (2025), our default reward combines a format reward $R_{\text{format}}$ and an accuracy reward $R_{\text{accuracy}}$. We *extend* the format reward so that, in addition to producing a reasoning segment and a verifiable answer, the model must also emit explicit ToM-role annotations. Concretely, $R_{\text{format}} = 1$ iff the output contains valid `<think></think>` and `<answer></answer>` tags *and* (for ToM items) includes, for each required order $k \in \{1, \ldots, K\}$, a pair of role tags `<TARGET_OF_MIND></TARGET_OF_MIND>` and `<TARGET></TARGET>` with non-empty content; otherwise $R_{\text{format}} = 0$. The accuracy reward evaluates the content in the `<answer></answer>` block, assigning $R_{\text{accuracy}} = 1$ if it exactly matches the ground truth and 0 otherwise.

In addition, we introduce a ToM-specific reward $R_{\text{tom}}$ that directly evaluates role attribution. Let $a_k^\star$ and $t_k^\star$ denote the ground-truth agent-of-mind and target at order $k$, and let $\hat{a}_k$ and $\hat{t}_k$ be the model's corresponding predictions extracted from the `<TARGET_OF_MIND></TARGET_OF_MIND>` and `<TARGET></TARGET>` tags, respectively. We define a simple binary scorer:

$$R_{\text{tom}} \;=\; \mathbb{1}\!\left[\bigwedge_{k=1}^{K} \left(\hat{a}_k = a_k^\star \;\wedge\; \hat{t}_k = t_k^\star\right)\right],$$

which returns 1 iff *all* predicted role pairs match the ground-truth belief hierarchy at every order (and 0 otherwise). We specifically enforce this to only be rewarded when all orders' predictions are correct because the results of messing up any order are equally unhealthy. This reward directly penalizes the characteristic ToM failure mode of mixing up the target and the target of mind. In total, the total new reward $R$ is defined as $R_{\textbf{accuracy}} + R_{\textbf{format}} + R_{\textbf{tom}}$ The results are shown in Table 10. As we can see, on OpenToM and ToMATO, our ToM rewards $R_{\text{tom}}$ reaches 90.56% and 91.92% respectively, signaling a 1.12% and 1.92% improvement, respectively.

# I  ADDITIONAL RL OPTIMIZATION ALGORITHMS

Besides the default GRPO, we have also experimented with two recently proposed algorithms DAPO (Yu et al., 2025) and GSPO (Zheng et al., 2025). We use standard settings otherwise and show results in Table 11. As we can see, both of these algorithms actually yield small ( 1%) improvement over the GRPO baseline. For example, on OpenToM, DAPO reaches 90.84%, 1.7% higher than GRPO on the same dataset. Nonetheless, GRPO suffices for the purposes of our study, given that they are all in the family of reinforcement learning with verifiable rewards.

Table 11: Comparison of DAPO and GSPO on **OpenToM** and **ToMATO**.

| Method | OpenToM | | ToMATO | |
|---|---|---|---|---|
| | DAPO | GSPO | DAPO | GSPO |
| Accuracy (%) | 90.84 | 89.66 | 91.12 | 90.10 |

# J  ADDITIONAL ANALYSIS

## J.1  DETAILS ON REASONING TRACE ANALYSIS AND ERROR ANALYSIS

In this section, we provide implementation details for the quantitative and qualitative analyses over reasoning traces. First, in the *Reasoning Trace + Zero-shot* setting, we freeze the original base model (Qwen2.5-7B-Instruct) and supply it with only the thinking traces generated by our Thinking-RFT model, with the final `<answer>` span removed. For each example, we prepend the original story and question, append the RFT `<think>` content as an auxiliary "hint", and then ask the base model to answer again with a single letter or short phrase. We score this setting identically to our main experiments. This measures whether the RFT-induced chains-of-thought encode task-relevant ToM structure that can improve the performance of a separately run zero-shot model, rather than being post hoc rationalizations.

Second, for the *LLM-as-a-judge* analysis, we collect reasoning traces from both Zero-shot and Thinking-RFT models on OPENTOM and TOMATO and prompt a stronger LLM to rate each trace along three axes: (i) logical consistency (LC: internal coherence and absence of contradictions), (ii) faithfulness (F: whether the reasoning faithfully reflects the given story and question, without hallucinating unsupported events), and (iii) efficiency (E: whether the reasoning is concise and focused on the causal hinge, rather than verbose or meandering). The judge operates on anonymized traces without access to model identities, and we report the average scores across all evaluated examples in Table 9.

Finally, for *error analysis*, we manually inspect the Thinking-RFT model's failures on OPENTOM and TOMATO with three PhD stuents. They independently label each error according to prede-fined categories (e.g., collapsing second-order queries into first-order reasoning, mis-tracking the

Table 12: LLM-judge scores for reasoning traces on **Explore_ToM** (shortcut-prone). "LC" = logical consistency, "F" = faithfulness, and "E" = efficiency. Scores are reported out of 10.

| Method | LLM-Judge (LC/F/E) |
| --- | --- |
| Zero-shot | 4.5 / 3.2 / 8.0 |
| Thinking-RFT (w/ shortcut) | 1.0 / 9.8 / 8.2 |

target agent or object, and incorrect final option despite largely correct intermediate reasoning), and we keep only labels on which at least two annotators agree. This yields a consensus view of the dominant failure modes and ensures that our qualitative conclusions are fair.

### J.2 REASONING TRACE ANALYSIS OVER TRAINING ON SHORTCUT DATASET.

Table 12 shows that when we train Thinking-RFT directly on the shortcut-prone Explore_ToM dataset, the *quality* of the reasoning suffer detrimentally. Compared to the zero-shot model, shortcut-tuned Thinking-RFT collapses logical consistency from 4.5 to 1.0, implying there is hardly any correct logic or ToM reasoning. The high faithfulness and efficiency are meaningless in this case as models overfitted to shortcut naturally only require a few tokens to arrive at an answer and do not need to fabricate any fact, as the answer is readily available (shortcut). This supports our claim that shortcut datasets not only bias quantitative comparisons, but also actively *degrade* the causal quality of reasoning traces, making the model appear grounded while its underlying inference is logically fragile.

### J.3 QUANTITATIVE STUDY OVER ATTENTION MAP OBSERVATIONS

Beyond the single example in Figure 5, we quantitatively test whether attention under Thinking-RFT focuses on the same sentences that humans regard as causally decisive. We created a 100-example subset(50 from OPENTOM, 50 from TOMATO) and hired three PhD students to independently mark the minimal set of *anchor sentences* whose content is sufficient to determine the correct answer, keeping anchors on which at least two annotators agree. For each story, we then rank input sentences by their average last-layer attention when the model generates its reasoning and final answer, and check whether *all* annotated anchors are contained among the top three attended sentences (anchors are typically 1–3 sentences long; if it is more than 3, then at least 3 need to be captured). Under this metric, the Thinking-RFT model covers the human-identified anchors in 89% of cases, indicating that its attention during reasoning is tightly concentrated on the minimal causal cues that humans themselves rely on to solve the task.

## K ADDITIONAL EXPERIMENTAL RESULTS

In this section we present additional experimental results. We first provide more attention visualizations and then provide other results.

### K.1 ADDITIONAL ATTENTION VISUALIZATION.

Here we provide 5 additional examples.

**Example 1 (Figure 6).** In this narrative, Sophie and Frank discover an onion in a box in the sunroom. Sophie then exits the room, after which Frank moves the onion to a more prominent spot. The question asks: *"From Sophie's perspective, is the onion in its initial location by the end of the story?"*—so the correct answer hinges on two key facts: (i) Sophie saw the onion in the box before leaving, and (ii) she does *not* observe Frank's relocation, so her belief remains that the onion is still in the box.

Figure 6 shows the last-layer attention of the Thinking-RFT model. Along the $x$-axis, each column corresponds to one input sentence (labeled by its first word), and along the $y$-axis we index the generated tokens in the reasoning and answer. In the left panel (reasoning tokens), the model places consistently high attention on the sentences describing Sophie exiting the sunroom and Frank later

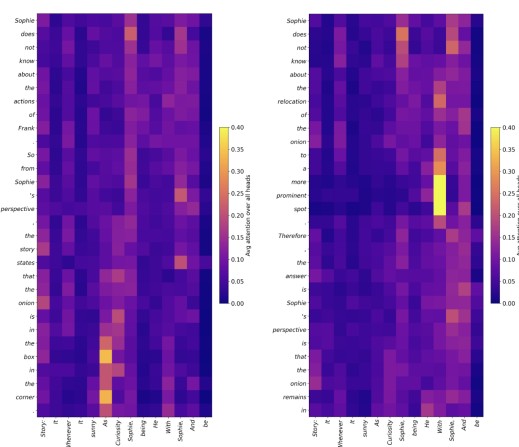

Figure 6: Example 1

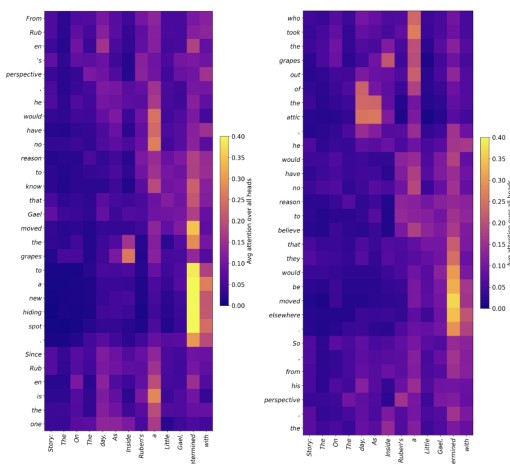

Figure 7: Example 2

moving the onion—exactly the causal hinge needed to infer Sophie's false belief. In the right panel (answer tokens), attention again concentrates on these same anchor sentences when producing the conclusion *"the onion remains in its initial location"*. This pattern illustrates how Thinking-RFT trains the model to focus its reasoning on the minimal belief-relevant events, rather than uniformly attending to all narrative details.

**Example 2 (Figure 7).** In this story, Gael hates grapes whereas Ruben loves them. Both agents see the grapes in an envelope in the attic; Ruben then leaves, and only *after* his departure does Gael secretly move the grapes to a new hiding spot. The question asks: "From *Ruben*'s perspective, are the grapes in their initial location by the end of the story?"—so the correct answer hinges on two key sentences: (i) the initial joint observation of the grapes ("Inside it lay a bunch of grapes, glistening with ripeness."), and (ii) Gael moving the grapes after Ruben has exited ("Gael swiftly moved the grapes to a new hiding spot...").

In the Thinking-RFT attention maps for this example, the model's chain-of-thought and final answer tokens place consistently high attention on the sentence columns labeled by the first words *Inside* and *Gael* (the two anchor sentences), forming clear vertical bands there, while assigning relatively low mass to surrounding descriptive sentences about preferences or atmosphere. This pattern shows that the RFT-trained model is explicitly tying together the "where Ruben last saw the grapes" event and the "Gael moves them unseen" event in order to answer the question about Ruben's belief, rather than diffusely attending to irrelevant parts of the story.

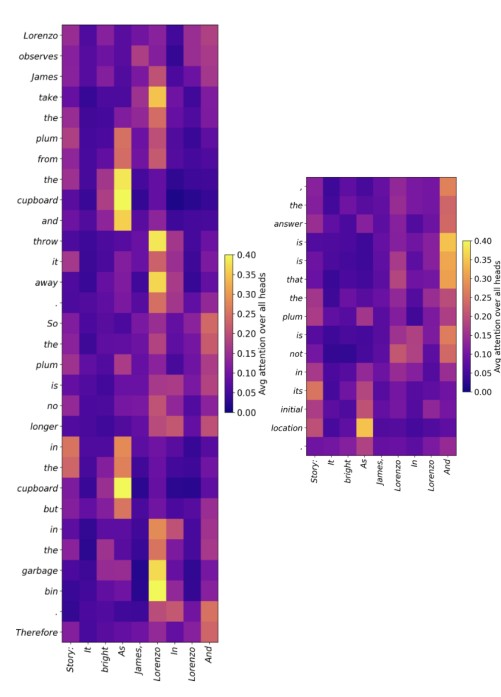

Figure 8: Example 4

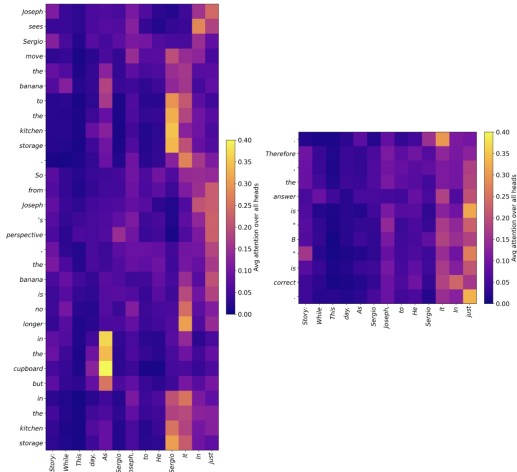

Figure 9: Example 3

**Example 3: Second-order belief Figure 9**    In this story, Sergio hates bananas while Joseph likes them. Both see a banana in the bedroom cupboard; Joseph briefly leaves, Sergio moves the banana to the kitchen storage, and Joseph re-enters *while witnessing* Sergio's action. The question asks: *"From Joseph's perspective, does Sergio think that the banana is in its initial location?"* Correct second-order ToM inference hinges on two anchor sentences: (i) "Sergio reached for the banana and carefully moved it to the kitchen storage," and (ii) "In that very moment, Joseph re-entered the room, witnessing Sergio's swift and purposeful action." In the attention maps for the Thinking-RFT model, the rows corresponding to the crucial reasoning tokens (e.g., "Joseph sees Sergio move the banana to the kitchen storage") and the final answer token place concentrated mass on the columns for these two anchor sentences, forming clear vertical bands over the segments labeled *As Sergio* and *In just*. This pattern indicates that the model is explicitly using the causal hinge—Joseph observing Sergio's relocation of the banana—to justify the (correct) conclusion that Joseph believes Sergio *does not* think the banana remains in its original location.

**Example 4: Second-order location Figure 8.**    This story describes James, who dislikes plums, and Lorenzo, who likes them. Both enter the kitchen and see a plum in the cupboard; James then picks up the plum and throws it into the garbage bin while Lorenzo watches. The question asks: *From Lorenzo's perspective, does James think that the plum is in its initial location by the end of the story?* Correct second-order ToM requires Lorenzo to reason that, because he *observes James personally remove the plum*, he knows that James is fully aware that the plum is no longer in the cupboard, so the answer is "No" (B).

In the attention maps, the RFT model focuses precisely on these causal hinge sentences. In the reasoning panel, the highest attention mass forms vertical bands over the sentences beginning with *"James take the plum from the cupboard"* and *"and throw it away ... in the garbage bin"*, which encode both the relocation and the fact that James himself performs the action. In the answer panel, the model again concentrates on the same columns when generating *"the plum is not in its initial location"*. This pattern indicates that, even for second-order questions, the RFT model grounds its judgment in the minimal evidence that connects James's observed action to his resulting belief.

## K.2    OTHER RESULTS.

Figure 11 presents an additional view of the results in Table 3. Table 13 presetns a fine-grained view of the results in Table 16.

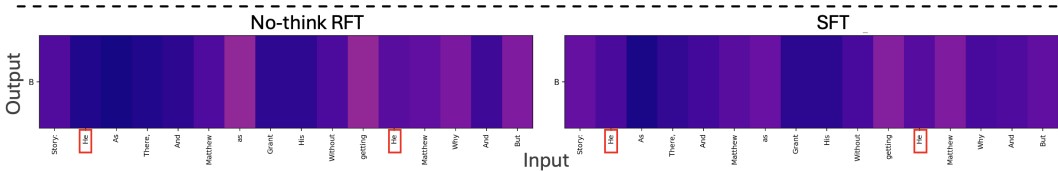

Figure 10: Attention visualization of No-Thinking-RFT vs. SFT. Neither baseline effectively attends to the critical causal tokens, reflecting weaker ToM reasoning.

Table 13: Generalization from 1st- to 2nd-order ToM on **OpenToM**. Best results are **bolded**.

| Method | First Order (*In-domain*) | | Multi-hop | Avg | Second Order (*OOD*) | | Multi-hop | Avg | Overall |
|---|---|---|---|---|---|---|---|---|---|
| | Location | | | | Location | | | | |
| | Cg | Fg | | | Cg | Fg | | | |
| Zero-shot | 49.00 | 48.00 | 55.00 | 50.67 | 51.00 | 33.00 | 53.00 | 45.67 | 46.43 |
| SFT | 97.00 | 93.00 | 89.00 | 93.00 | 76.00 | 44.00 | 76.00 | 65.33 | 79.17 |
| RFT | 99.00 | 95.00 | 91.00 | 95.00 | **84.00** | **59.00** | **80.00** | **74.33** | **84.67** |

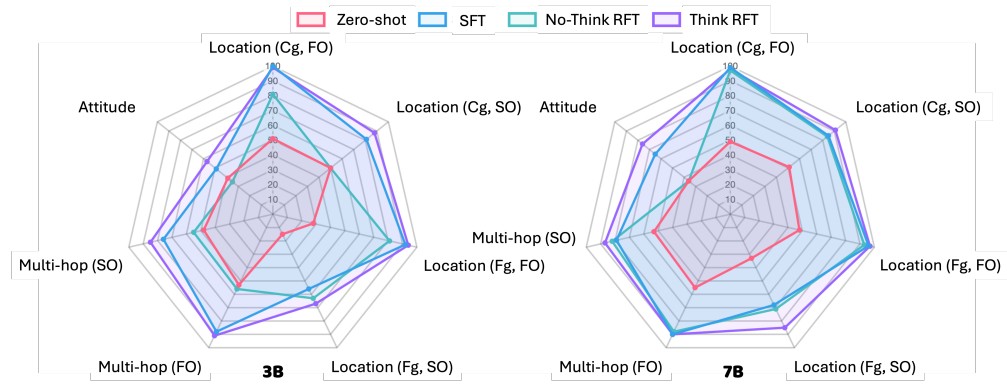

Figure 11: Radar figure of performance of different models on OpenToM.

Table 14: Performance on **ToMATO** (1st-order). Results are reported across five ToM categories with micro and macro averages.

| Method | Belief | Desire | Emotion | Intention | Knowledge | Avg |
|---|---|---|---|---|---|---|
| **First Order** (In-domain) | | | | | | |
| Zero-shot | 72.12 | 78.49 | 71.57 | 69.92 | 72.51 | 72.96 |
| 7B SFT | 87.61 | 88.17 | 87.25 | 90.65 | 86.73 | 88.08 |
| 7B RFT | 87.61 | 91.40 | 86.76 | 92.68 | 88.15 | 89.32 |
| **Second Order** (OOD) | | | | | | |
| Zero-shot | 59.29 | 66.39 | 64.25 | 60.61 | 60.56 | 62.22 |
| 7B SFT | 83.63 | 85.66 | 82.61 | 80.30 | 76.53 | 81.74 |
| 7B RFT | 84.51 | 85.66 | 82.61 | 83.33 | 87.79 | 84.78 |

Table 15: RFT outperforms No-Thinking-RFT and SFT on **ToMATO (conversational)** (Shinoda et al., 2025) across major categories and average. The best results in each column are highlighted.

| Method | Belief | Desire | Emotion | Intention | Knowledge | Avg$_\Delta$ vs. SFT |
|---|---|---|---|---|---|---|
| Zero-shot | 65.71 | 71.63 | 67.88 | 65.77 | 66.51 | 67.50$_{\downarrow 20.42}$ |
| SFT | 87.17 | 89.07 | 90.05 | 86.94 | 87.05 | 87.92 |
| No-Thinking RFT | 85.18 | 89.53 | 84.91 | 87.61 | 88.21 | 87.09$_{\downarrow 0.83}$ |
| Thinking-RFT | 88.50 | 92.33 | 88.81 | 90.54 | 89.86 | 90.00$_{\uparrow 2.08}$ |

Table 16: Generalization to unseen environments on **OpenToM**. Results are reported on one seen and three unseen environments, with an overall average.

| Method | **Seen** | $\hookrightarrow$**Unseen** | | | Avg |
|---|---|---|---|---|---|
| | Household | Medical / Hospital | Museum | Airport / Cabin | |
| Zero-shot | 46.43 | 45.33 | 47.00 | 46.66 | 46.33 |
| SFT | 83.14 | 75.33$_{\downarrow 7.81}$ | 76.00$_{\downarrow 7.14}$ | 76.66$_{\downarrow 6.48}$ | 75.99$_{\downarrow 7.15}$ |
| RFT | **89.14** | 87.67$_{\downarrow 1.47}$ | 88.66$_{\downarrow 0.48}$ | 88.33$_{\downarrow 0.81}$ | 88.22$_{\downarrow 0.92}$ |

