# OpenReview forum: "From Shortcuts to Reasoning: Robust Post-Training of Theory of Mind with Reinforcement Learning"
_ICLR.cc/2026/Conference — Submitted to ICLR 2026_

### Official Review · Reviewer_c2tS · 2025-10-26

**Soundness:** 2
**Presentation:** 2
**Contribution:** 3
**Rating:** 4
**Confidence:** 4

**Summary:**

This work investigates potential shortcuts in existing ToM datasets, and train LMs via RL on the shortcut-free datasets.

The main findings are as follows:

(1) several datasets are prone to shortcut exploitation and are therefore unsuitable for training;

(2) when trained on shortcut-free datasets, the performance follows the order thinking-RFT > no-thinking-RFT > SFT;

(3) thinking-RFT exhibits strong generalization to unseen domains and higher-order ToM questions; and

(4) RFT improves performance by learning to ground its reasoning in cues that correspond to causal factors.

Overall, the paper attempts to address an interesting question, but it lacks important methodological details and sufficient evidence to support its claims, especially (1) and (4). The writing is generally clear, though there are some typos. I will outline my specific concerns below.

**Strengths:**

- The writing is mostly clear and easy to follow.

- A major contribution of this work lies in its attempt to identify and analyze shortcuts in existing ToM datasets. Training on such datasets may encourage models to exploit superficial correlations rather than develop genuine ToM capabilities.

- The finding that RFT training outperforms SFT is not entirely surprising, since similar trends have been observed in other domains. However, this work reaches different conclusions from Lu et al. (2025) that explored similar questions.

- I appreciate the use of mechanistic analyses to investigate where RFT demonstrates its advantages, although the presented evidence is not fully convincing.

Reference:
Lu, Y. L., Zhang, C., Song, J., Fan, L., & Wang, W. (2025). Do Theory of Mind Benchmarks Need Explicit Human-like Reasoning in Language Models?. arXiv preprint arXiv:2504.01698.

**Weaknesses:**

- The paper reaches a different conclusion from Lu et al. (2025) but does not discuss the differences in detail. It seems the authors suggest that Lu et al. (2025) trained on shortcut-prone datasets, which may explain their opposite findings. Please provide a more detailed comparison and discussion in the related work section to clarify the source of these discrepancies.

- Some terms and methods are not clearly illustrated.
  - In Section 2.1, what exactly is the “stratified seed set”? For example, does it include (x, y) pairs where x is a question and y is the answer? How is the set “stratified”?
  - How are the heuristics implemented and combined? These parts should be described more clearly for reproducibility and clarity, since it's suppose to support a crucial claim of this work.

- Not enough evidence to support the claims:

  - The paper claims that training on shortcut datasets can harm ToM abilities, but there isn’t enough evidence to support this. It would be more convincing if the authors trained the same model on these shortcut datasets using the same method and compared the results, instead of only showing a qualitative figure with reasoning-trace errors (Figure 2).

  - Using procedurally generated data is not very convincing for testing generalization (e.g., generating data from 'apartment' to unseen 'outer space' for MMToM), since such data still follow similar logic as the training data. It would be stronger to test on truly out-of-domain datasets, e.g., the shortcut-prone datasets mentioned earlier, to see if RL really improves generalizable ToM abilities.

- Typos for revising the manuscript:
  - Line 118: casual -> causal
  - Line 152: there should be '.' before 'On'
  - Line 155: mentioned section 2 -> mentioned in Section 2
  - Line 159: Table 2 -> Figure 2

**Questions:**

- Regarding generalization, could you test on existing evaluation datasets instead of constructing new ones derived from the training data? This would make the generalization claim more convincing.

- In Section 5.2, you mention that 'we manually mark the minimal cues in each narrative that establish the causal hinge between agents’ intentions and outcomes.' This appears to suggest a quantitative evaluation across multiple data, yet the paper only provides a single qualitative example. Please include quantitative results or clarify how the experiments were truly performed to better support the crucial claim of 'RFT improves performance by learning to ground its reasoning in cues that correspond to causal factors'.

**Details Of Ethics Concerns:**

No ethics concerns.

---

> ### Author Response · Authors · 2025-11-22
> **Rebuttal Part 1**
>
> We would like to thank you for your diligent efforts and constructive suggestions on our paper, which have helped us think more deeply. We are also glad you recognized our mechanistic analyses. In the following, we have provided a point-by-point response to all questions raised.
>
>
> >W1. The paper reaches a different conclusion from Lu et al. (2025) but does not discuss the differences in detail. It seems the authors suggest that Lu et al. (2025) trained on shortcut-prone datasets, which may explain their opposite findings. Please provide a more detailed comparison and discussion in the related work section to clarify the source of these discrepancies.
>
> **R1.** Thank you for pointing this out. We definitely recognized the importance and value of the mentioned work and have revised the part discussing these two works in the related work into the following (Sec. 6): Closest to our work is Lu et al. [1], who also study ToM post-training with RL and SFT but arrive at different conclusions: on shortcut-prone benchmarks (Hi-ToM, Explore-ToM) they report that both RL and SFT quickly exceed 90% accuracy and SFT outperforms RFT. This matches our own observations in our shortcut experiment (Sec. 2.3). In contrast, our work first audits ToM datasets to identify and remove shortcut-heavy ones, and then shows that on cleaned ToM suite Thinking-RFT consistently outperforms SFT and No-Thinking-RFT in accuracy and generalization. Together, these results suggest that the discrepancy is largely explained by differences in benchmark design and dataset selection, and highlight the importance of careful data filtering and shortcut auditing when conducting reasoning-focused post-training.
>
> ---
>
>
> >W2 Some terms and methods are not clearly illustrated. In Section 2.1, what exactly is the “stratified seed set”? For example, does it include (x, y) pairs where x is a question and y is the answer? How is the set “stratified”? ...
>
> **R2.1** We appreciate the reviewer's attention to detail and thank you for raising this valid concern. We apologize for not including enough detail originally due to space — we have clarified it below and also provied detailed explanations in Appendix E in the revision.
>
> In our shortcut-auditing framework, the “stratified seed set” D_seed is a small labeled subset of each dataset consisting of full (x, y) QA pairs, where x = story + question + options and y = correct answer (letter for multiple choice or the answer token for free-form questions). We construct D_seed by sampling examples uniformly across ToM category (e.g., belief / location / attitude), ToM order (1st / 2nd / 3rd / 4th where available), and answer label. Essentially, **stratified** means the seed set is **balanced over the main structural axes of the dataset**, so the agent sees a **representative mix of problem types instead of a skewed slice**.
>
>
> >W2. ... How are the heuristics implemented and combined? These parts should be described more clearly for reproducibility and clarity, since it's suppose to support a crucial claim of this work.
>
>
> **R2.2** This is a good question. In our experiment, each heuristic is manually implemented in a **Python script** following all rules and steps in a zero-update manner (no learning, model involvement; implementation takes about 2-20 minutes for one with proficient Python programming); this ensures that it can be executed fast. Next, we do not combine heuristics into an ensemble: each candidate shortcut is evaluated **independently**, and for each dataset we retain the single accepted heuristic with the highest accuracy as its shortcut proxy; this ensures that our audit is **conservative and minimizes false negatives**. We repeat this procedure multiple times and flag a dataset as ``shortcut-prone'' if any such heuristic meets the threshold.

---

> > ### Author Response · Authors · 2025-11-22
> > **Rebuttal Part 2**
> >
> > >W3. Not enough evidence to support the claims: The paper claims that training on shortcut datasets can harm ToM abilities, but there isn’t enough evidence to support this. It would be more convincing if the authors trained the same model on these shortcut datasets using the same method and compared the results, instead of only showing a qualitative figure with reasoning-trace errors (Figure 2).
> >
> > **R3.1** This is a very good point, we agree with the reviewer and greatly appreciate the feedback. In the revision, we added an explicit **shortcut-training experiment** (Sec. 2.3, Tab. 2): we fine-tune 3B and 7B models with SFT, Thinking-RFT, and No-Thinking-RFT on the shortcut-prone Explore-ToM dataset, and evaluate both in-domain (Explore_ToM) and out-of-domain on Hi-ToM. We note that both benchmarks are flagged as shortcut-heavy by our audit (Tab. 1, Sec. 2.2). The new results are reported in the table below
> >
> > | Method            | 3B In-domain | 3B OOD | 7B In-domain | 7B OOD |
> > |-------------------|-------------:|-------:|-------------:|-------:|
> > | Zero-shot         | 49.5         | 38.5   | 62.0         | 43.6   |
> > | SFT               | 96.4         | 32.5   | 95.8         | 34.2   |
> > | Thinking-RFT      | 93.2         | 31.3   | 94.3         | 35.3   |
> > | No-Thinking RFT   | 95.8         | 32.0   | 96.1         | 34.0   |
> >
> > These numbers empirically reveal **4 drawback** of shortcut-prone data:
> >
> > 1. **Inverted ranking of post-training methods.**
> >    On Explore-ToM, the apparent hierarchy is SFT ≥ No-Thinking-RFT ≥ Thinking-RFT (all ≈93–96%), which is the **reverse** of our shortcut-free setting where Thinking-RFT > SFT > No-Thinking-RFT across four datasets. This illustrates that shortcuts hinder a faithful comparison of post-training methods and the misleading ranking would cause practitioners to prefer the wrong post-training strategy for enhancing ToM.
> >
> > 2. **Scaling effects are erased.**
> >    On clean OpenToM, increasing model size from 3B → 7B yields consistent +5–6 point gains for both SFT and Thinking-RFT. Under shortcut training, 3B and 7B collapse to almost identical in-domain performance (≈93–96% for all methods), and 3B SFT even slightly surpasses 7B SFT (96.4% vs 95.8%). We hypothesize that this is because overfitting cheap heuristics do not require much capacity. This also means that when trained on shortcut data, additional capacity brings no benefit and most capacity is wasted.
> >
> > 3. **Generalization is actively harmed (negative transfer).**
> >    For both 3B and 7B, tuning on Explore_ToM boosts in-domain accuracy by +40–45 points over zero-shot (e.g., 49.5 → 96.4 for 3B SFT), yet failes to generalize: on Hi-ToM performance *drops* by around −6–9 points (e.g., 38.5 → 31–33 for 3B; 43.6 → 34–35 for 7B). This is also the **opposite** of our cross-dataset genearlization results in Table 6 where RFT successfully improve performance even cross-dataset. This shows that shortcut data inflates in-domain accuracy and only induces a false sense of ToM.
> >
> > 4. **Shortcut training teaches no genuine ToM reasoning.**
> >    Complementing the accuracy numbers, we also perform the LLM-as-judge analysis and report the result below (also in Appendix H Table 10). The result shows that reasoning traces from shortcut-trained models score especially low in logical coherence (1/10), much lower than the Zero-shot+CoT baseline. The faithfulness metric is high because the shortcuts themselves are facts within the text, hence learning on them wouldn't decrease its score. However, the problematic logical reasoning proves that training on shortcuts does not result in genuine ToM reasoning.
> >
> > | Method                      | LLM-Judge (LC / F / E) |
> > |-----------------------------|--------------------------|
> > | Zero-shot                   | 4.5 / 3.2 / 8.0          |
> > | Thinking-RFT (w/ shortcut) | 1.0 / 9.8 / 8.2          |
> >
> >
> > So to sum up, these new experiments support our conceptual argument with direct empirical evidence: shortcut-prone datasets not only fail to reveal the benefits of RFT, they can invert method rankings, erase scaling effects, and even induce negative transfer.

---

> ### Author Response · Authors · 2025-11-22
> **Rebuttal Part 3**
>
> >W3. Not enough evidence to support the claims: Using procedurally generated data is not very convincing for testing generalization (e.g., generating data from 'apartment' to unseen 'outer space' for MMToM), since such data still follow similar logic as the training data. It would be stronger to test on truly out-of-domain datasets, e.g., the shortcut-prone datasets mentioned earlier, to see if RL really improves generalizable ToM abilities.
>
> **R3.2** Thank you for raising this important point. We agree that robust reasoning transferability is vital. We have added this experiment below and reflected the change in Sec. 4.3 and Table 6 in the revision. Here we evaluate the model trained on OpenToM on OOD datasets ToMATO [1], Explore-ToM [2], and Hi-ToM [3]. We use OpenToM as the source domain since it is the only one that contains both free-form and multiple choice format questions, which are used exclusively by other datasets. We do note that there are still substantial domain shift, for example, OpenToM focus on belief&knowledge questions while ToMATO focus on desire&intention questions, which calls for a diffrent set of skills to solve effectively.
>
>
>
> | Method | Train    | Eval                     | Acc   |
> |--------|----------|---------------------------|-------|
> | 0-shot | –        | **ToMATO**                    | 67.5  |
> | RFT    | OpenToM  | ToMATO                    | 70.4  |
> | SFT    | OpenToM  | ToMATO                    | 56.8  |
> |--------|----------|---------------------------|-------|
> | 0-shot | –        | **Explore-ToM**               | 62.0  |
> | RFT    | OpenToM  | Explore-ToM               | 71.0  |
> | SFT    | OpenToM  | Explore-ToM               | 64.5  |
> |--------|----------|---------------------------|-------|
> | 0-shot | –        | **Hi-ToM (1&2 order)**         | 55.0  |
> | RFT    | OpenToM  | Hi-ToM (1&2 order)         | 60.0  |
> | SFT    | OpenToM  | Hi-ToM (1&2 order)         | 48.3  |
>
>
> As shown in the table, across cross-dataset experiments, RFT is the only method that **consistently improves** performance on both OOD datasets: accuracy rises from 67.5% to 70.4% on ToMATO, from 62.0% to 71.0% on Explore-ToM, and from 55% to 60% on Hi-ToM; SFT produces mixed changes but are mostly negative; notebaly, on ToMATO and Hi-ToM, it exchibits **significant drop** (-10.7% and -16.7%). This indicates that the ToM skills acquired via Thinking-RFT on a single shortcut-free dataset remain robust under substantially distribution shift, whereas SFT offers limited and inconsistent transfer.
>
>
> ---
>
> >W4. Typos for revising the manuscript:
>
> **R4.** Thank you so much for catching these. We apologize for the negligence and have fixed them in the updated manuscript. (In original ducoment) Line 118: casual → causal, Line 152: added missing period before “On”, Line 155: “mentioned section 2” → “mentioned in Section 2", and Line 159: Table 2 → Figure 2.
>
> ---
>
> >Q1. Regarding generalization, could you test on existing evaluation datasets instead of constructing new ones derived from the training data? This would make the generalization claim more convincing.
>
> **A1.** Please refer to **R3.2** above.
>
> >Q2. In Section 5.2, you mention that 'we manually mark the minimal cues in each narrative that establish the causal hinge between agents’ intentions and outcomes.' This appears to suggest a quantitative evaluation across multiple data, yet the paper only provides a single qualitative example. Please include quantitative results or clarify how the experiments were truly performed to better support the crucial claim of 'RFT improves performance by learning to ground its reasoning in cues that correspond to causal factors'.
>
> **A2.** To further support the causally coherent reasoning claim, we performed a small quantitative analysis in Appendix J.3 in the revision. We hired **3 PhD Students** and manually annotated a test set of 200 examples from OpenToM and ToMATO with anchor sentences (tokens that contain causal cues). We extract the top 5 most attended sentences and evaluate how often they contain the anchor sentences; the results in the table below show that Thinking-RFT captures anchor sentences **significantly more** often (**89%**) than the zero-shot model (**32%**). We have reflected these results in the updated PDF. Additionally, we have also **added more qualitative examples** to Appendix J in the updated PDF.
>
> **References**
>
> [1] ToMATO: Verbalizing the Mental States of Role-Playing LLMs for Benchmarking Theory of Mind
>
> [2] Explore Theory of Mind: Program-guided adversarial data generation for theory of mind reasoning
>
> [3] HI-TOM: A Benchmark for Evaluating Higher-Order Theory of Mind Reasoning in Large Language Models

---

> ### Comment · Reviewer_c2tS · 2025-11-25
> **Raised score**
>
> I think the additional experiments address the main weaknesses of the original manuscript. Please continue work on presentation and clarifying the important details.

---

> > ### Author Response · Authors · 2025-11-26
> > **Thank you for reviewing our paper**
> >
> > We sincerely thank you again for your detailed review and constructive feedback and for increasing the score. We are glad that our additional results addressed your main concerns — we will for sure continue work on the presentation and discuss the important details thoroughly in the final version!

---

### Official Review · Reviewer_MqzA · 2025-10-27

**Soundness:** 2
**Presentation:** 3
**Contribution:** 2
**Rating:** 4
**Confidence:** 4

**Summary:**

This paper aims to tackle the generalizability and counterfactual robustness of foundation models on theory-of-mind (ToM) reasoning tasks. They first design a principled filtering approach that identifies the shortcut issue in the existing ToM benchmarks, especially for the datasets like Hi-ToM higher-order queries. After identifying the shortcut-free dataset, the authors further make a comprehensive comparison among the performance of zero-shot and different post-training approaches, including SFT and RFT (with or without thinking tokens). The results demonstrate that RFT with thinking outperforms other post-training approaches in most tasks. RFT enjoys not only the best accuracy on different ToM benchmarks, but also better generalizability to higher-order reasoning cases and better causal consistency to counterfactual probing. The qualitative studies on the attention map also reveal that the model after RFT is more capable of capturing the semantics in the contexts.

**Strengths:**

* The paper is generally well-written and easy to follow. The clarity on the experiment settings is good.
* The shortcut perspective on the current ToM benchmark is interesting and worth studying deeper for the ToM research community.
* The empirical results on different post-training approaches, as well as the qualitative studies are clearly presented and can justify the major claim of contribution in this paper.

**Weaknesses:**

* **Incoherent contribution in section 2 and 3:** It's a little bit unclear what is the necessary logical connection between section 2 and 3. Section 2 only identifies those dataset that potentially has confounded shortcuts that should be precluded from post-training. However, these datasets with shortcut are also held away from evaluation set in the final experiments with generalization.
* **Missing baselines:** There exists quite a few baselines in solving ToM-QA tasks, such as SimToM [1], AutoToM [2]. However, the paper misses these baselines that are currently leading approaches in the MMToM benchmark.
* **Limited analysis on the higher-order generalizability:** Since the authors only use shortcut-free dataset like OpenToM to conduct post-training and evaluation, the claim on 'generalizability to higher-order queries' does not seem to be sufficiently supported as they only care about the generalization from first-order to second-order ToM. A more comprehensive studies on third-order and fourth-order ToM (like the ones in Hi-ToM) will be interesting.
* **Missing evaluation on cross-dataset generalizability:** It is hard to justify whether the results the authors present are coming from better overfitting to a single dataset, or indeed a better capability in general reasoning across different ToM datasets. Therefore, it is necessary to add the evaluation on different ToM datasets after the SFT/RFT, rather than a small test split which is less different in the QA domain.
* **Limited scale of evaluation:** The OpenToM dataset, the authors only select 100 samples from each category as evaluation. They also missed the evaluation on the test split provided by the MMToM leaderboard. These limits the contribution of the proposed method.
* **Limited scale of qualitative analysis**: The author only demonstrates one pair of qualitative comparisons on the attention map. More qualitative examples can be provided in the appendix to make the claim of causality-coherent reasoning more solid.
* **Missing analysis on the 'spurious correlation'**: The authors mentioned their motivation comes from the observation that the model 'simply exploiting spurious correlations'. However, the analysis terminates after section 2 right after they preclude the Hi-COM and other datasets in the finetuning and evaluation dataset. It will be more reasonable if they can conduct **quantitative** analysis on how (a) the data filtering (judged by simple rules and lexical association), as well as (b) the SFT/RFT-style post-training, can help mitigate such spurious correlation exploitation, even on those benchmarks with shortcuts.

> [1] Wilf, Alex, et al. "Think twice: Perspective-taking improves large language models' theory-of-mind capabilities." ACL 2024.
>
> [2] Zhang, Zhining, et al. "Autotom: Automated bayesian inverse planning and model discovery for open-ended theory of mind." *ICLR 2025 Workshop on Foundation Models in the Wild*. 2025.

**Questions:**

See the weakness section for the main questions I have and the comments. I would consider re-adjust my assessment if the majority of them get resolved during the rebuttal phase.

---

> ### Author Response · Authors · 2025-11-22
> **Rebuttal Part 1**
>
> We genuinely appreciate that you recognized the shortcut issue we raised as important and "worth studying deeper", and our experiment was well-presented. Below we address your questions in detail.
>
>
> > W1. Incoherent contribution in section 2 and 3: It's a little bit unclear what is the necessary logical connection between section 2 and 3. Section 2 only identifies those dataset that potentially has confounded shortcuts that should be precluded from post-training.
>
> **R1.1** Thank you for the thoughtful question. Please let us clarify here. First, we want to reiterate the two main challenges our work is addressing:
>
> - **Internalizing ToM reasoning.** ToM reasoning is intrinsically difficult: it requires the model to reason across a long horizon while tracking different objects, characters, which current foundation models lack [1-3]. Reasoning-based methods have been successfully proposed and shown great improvement [1,4-6]; however, these approaches typically rely on intricate multi-step prompting framework [1,6,5] around a strong backbone (e.g., GPT-4o) [4,5], which increases inference-time complexity and makes deployment more challenging. Our goal is to see if such ToM-specific reasoning can be **directly learned** via RL (and if so, when and how), without the manual hurdles.
>
> - **Shortcut-prone dataset.** When trying to solve (i), we discovered this critical issue that surprisingly exists in many ToM datasets. We have performed detaile experiments in Sec. 2.2 to highlight the harm of shortcut in detail — most importantly, it prevents models from developing genuine reasoning (as ML models tend to find the easiest path[7]), which goes directly against the goal for (i) (learning true ToM reasoning).
>
> Given these two challenges our work first addresses the shortcut issue by proposing a lightweight yet efficient audit framework (Sec. 2.1), identifying and removing the shortcut-heavy ones (Sec 2.2), and then study the first challange of efficiently learning human-level ToM reasoning (Sec. 3-4). In essence, Section 2 sets a prerequisits for post-training study in Section 3; without the analysis in Section 2 (we added results on the harm of shortcut data in detail in Sec 2.3 in the revision), experiments in Section 3 would be misleading. We have clarified this in the revised introduction (Sec. 1) as well.
>
> ---
>
> >W1. ... However, these datasets with shortcut are also held away from evaluation set in the final experiments with generalization.
>
> **R1.2** This is a valid point. Originally, we did not include these datasets for evaluation since we did not evaluate OOD generalization across datasets as the distribution shift could be rather large (different focus, question formats, etc,. which would confound the generalization results with other factors). However, we agree with you that such an evaluation could be valuable, so we have added an evaluation on one of the shortcut-prone benchmarks as part of the newly added cross-dataset experiments in Sec 4.3 & Table 6 (please also refer to **R4** below).
>
> > W2. Missing baselines: There exists quite a few baselines in solving ToM-QA tasks, such as SimToM [1], AutoToM [2]. However, the paper misses these baselines that are currently leading approaches in the MMToM benchmark.
>
>
> **R2.** Thank you for the suggestion, and we apologize for not including these baselines originally. We have now cited and added both **SimToM** and **AutoToM** as new baselines in the table below, as well as in the updated version (Table 3 & 4), all using the same backbone as ours (Qwen2.5-7B family). On MMToM, SimToM and AutoToM yield +3.3% and +12.4% improvements respectively, both **substantially lower** than the gain achieved by our method (**+42.7%**). Moreover, our approach reaches **83.3%** accuracy on MMToM—higher even than AutoToM + GPT-4o (79.8%), despite GPT-4o being a much stronger base model.
>
>
> | Method                   | OpenToM | ToMATO | MMToM | MuMA-ToM |
> |--------------------------|---------|--------|-------|----------|
> | Zero-shot                | 46.4    | 67.5   | 45.0  | 43.3     |
> | SimToM                   | 49.7    | 69.3   | 50.6  | 49.6     |
> | AutoToM                  | 56.8    | 69.4   | 56.9  | 59.1     |
> | Thinking-RFT             | 89.1    | 90.0   | 83.3  | 81.1     |
>
>
> We kindly note that most ToM methods, including the two evaluated, are inference-time algorithms that do not update model weights, whereas ours (RFT post-training) does, thus they operate in a different regime. Also note that the original AutoToM paper employs GPT-4o as the backbone, hence lower performance is expected when switched to the weaker Qwen-7B model.

---

> > ### Author Response · Authors · 2025-11-22
> > **Rebuttal Part 2**
> >
> > > W3. Limited analysis on the higher-order generalizability: Since the authors only use shortcut-free dataset like OpenToM to conduct post-training and evaluation, the claim on 'generalizability to higher-order queries' does not seem to be sufficiently supported as they only care about the generalization from first-order to second-order ToM. A more comprehensive studies on third-order and fourth-order ToM (like the ones in Hi-ToM) will be interesting.
> >
> >
> > **R3.** We agree that evaluating higher-order ToM (3rd and 4th order) is valuable. However, the only public dataset that contains higher-order questions is Hi-ToM, and our shortcut analysis shows that this dataset **relies heavily on shortcut patterns**. Because of this, we cannot fine-tune on the 1st and 2nd order questions in Hi-ToM and then evaluate generalization on the 3rd and 4th order questions, since the shortcut issue would make the results unreliable.
> >
> >
> > Even so, we still evaluate our models trained on OpenToM (1st and 2nd order) directly on Hi-ToM’s 3rd and 4th order questions. We note that since this setting includes both major distribution shift and order shift, it would be less effective in accesing ToM skill transferability; we believe the cross-dataset generalization result in R4 below is a more reasonable evaluation. Nevertheless, the result is that on 3rd order questions, RFT reaches 40% (vs. 36.6% Zero-shot baseline) while SFT stays the same, and on 4th order questions, RFT reaches 23.3% (vs 20% Zero-shot baseline) while SFT drops to 18.8%. This shows that RFT still transfers better than SFT even in this challenging setting.
> >
> >
> > ---
> >
> >
> > > W4. Missing evaluation on cross-dataset generalizability: It is hard to justify whether the results the authors present are coming from better overfitting to a single dataset, or indeed a better capability in general reasoning across different ToM datasets. Therefore, it is necessary to add the evaluation on different ToM datasets after the SFT/RFT, rather than a small test split which is less different in the QA domain.
> >
> >
> > **R4.** Thank you for raising this important point. We agree that robust reasoning transferability is vital. We have added this experiment below and reflected the change in Sec. 4.3 and Table 6 in the revision. Here we evaluate the model trained on OpenToM on OOD datasets ToMATO [1], Explore-ToM [2], and Hi-ToM [3]. We use OpenToM as the source domain since it is the only one that contains both free-form and multiple choice format questions, which are used exclusively by other datasets. We note that there are still substantial domain shift, for example, OpenToM focus on belief&knowledge questions while ToMATO focus on desire&intention questions, which calls for a diffrent set of skills to solve effectively.
> >
> >
> >
> > | Method | Train    | Eval                     | Acc   |
> > |--------|----------|---------------------------|-------|
> > | 0-shot | –        | **ToMATO**                    | 67.5  |
> > | RFT    | OpenToM  | ToMATO                    | 70.4  |
> > | SFT    | OpenToM  | ToMATO                    | 56.8  |
> > |--------|----------|---------------------------|-------|
> > | 0-shot | –        | **Explore-ToM**               | 62.0  |
> > | RFT    | OpenToM  | Explore-ToM               | 71.0  |
> > | SFT    | OpenToM  | Explore-ToM               | 64.5  |
> > |--------|----------|---------------------------|-------|
> > | 0-shot | –        | **Hi-ToM (1&2 order)**         | 55.0  |
> > | RFT    | OpenToM  | Hi-ToM (1&2 order)         | 60.0  |
> > | SFT    | OpenToM  | Hi-ToM (1&2 order)         | 48.3  |
> >
> >
> > As shown in the table, across cross-dataset experiments, RFT is the only method that **consistently improves** performance on both OOD datasets: accuracy rises from 67.5% to 70.4% on ToMATO, from 62.0% to 71.0% on Explore-ToM, and from 55% to 60% on Hi-ToM; SFT produces mixed changes but are mostly negative; notebaly, on ToMATO and Hi-ToM, it exchibits **significant drop** (-10.7% and -16.7%). This indicates that the ToM skills acquired via Thinking-RFT on a single shortcut-free dataset remain robust under substantially distribution shift, whereas SFT offers limited and inconsistent transfer.

---

> > > ### Author Response · Authors · 2025-11-22
> > > **Rebuttal Part 3**
> > >
> > > > W5. Limited scale of evaluation: The OpenToM dataset, the authors only select 100 samples from each category as evaluation. They also missed the evaluation on the test split provided by the MMToM leaderboard. These limits the contribution of the proposed method.
> > >
> > >
> > > **R5.** This is a valid concern, and we apologize for the lack of clarity in the original draft. The full OpenToM dataset contains over 10K questions in total and over 1K questions per category. Our goal was to select a subset that is sufficiently large to provide **representative and statistically reliable evaluation**. Accordingly, our reported numbers are averaged over three runs, each using 100 questions per category. We believe this sample size offers a stable estimate of performance. To further address the reviewer’s concern, we additionally evaluated our model on the entire test set, and the resulting accuracy (**89.5%**) is **slightly higher** than the subset-based result (**89.14%**). This confirms that our sampling strategy does not inflate performance. We have added a clarification and the full-test evaluation details to Appendix E in the updated PDF.
> > >
> > >
> > > On MMToM, we agree with the reviewer that evaluating on the test set (40 videos/120 questions) would further strengthen the evidence. However, this split has **not been publicly released**. We checked the official GitHub repo, project page, and HuggingFace dataset and could only find the main benchmark (134 videos / 600 questions). We are more than **happy to evaluate** our model on the test set once it becomes available.
> > >
> > >
> > > ---
> > >
> > >
> > > > W6. Limited scale of qualitative analysis: The author only demonstrates one pair of qualitative comparisons on the attention map. More qualitative examples can be provided in the appendix to make the claim of causality-coherent reasoning more solid.
> > >
> > >
> > > **R6.** This is a good suggestion. We have **added more examples** to Appendix J in the updated PDF. To further support the causally coherent reasoning claim, we performed a small quantitative analysis in Appendix J.3 in the revision. We hired **3 PhD Students** and manually annotated a test set of 200 examples from OpenToM and ToMATO with anchor sentences (tokens that contain causal cues). We extract the top 5 most attended sentences and evaluate how often they contain the anchor sentences; the results in the table below show that Thinking-RFT captures anchor sentences significantly more often (**89%**) than the zero-shot model (**32%**). We have reflected these results in the updated PDF.

---

> > > > ### Author Response · Authors · 2025-11-22
> > > > **Rebuttal Part 4**
> > > >
> > > > > W7. Missing analysis on the 'spurious correlation': The authors mentioned their motivation comes from the observation that the model 'simply exploiting spurious correlations'. However, the analysis terminates after section 2 right after they preclude the Hi-COM and other datasets in the finetuning and evaluation dataset. It will be more reasonable if they can conduct quantitative analysis on how (a) the data filtering (judged by simple rules and lexical association) ....
> > > >
> > > >
> > > > **R7.1** Thank you for the insightful feedback. We demonstrate that our data filtering is effective primarily through showing that training on short-cut prone dataset reaches the **opposite** results as shortcut-free data obtained using our data filtering method. Concretely, we fine-tune 3B and 7B models with SFT, Thinking-RFT, and No-Thinking-RFT on the shortcut-prone Explore-ToM dataset, and evaluate both in-domain (Explore_ToM) and out-of-domain on Hi-ToM. We note that both benchmarks are flagged as shortcut-heavy by our audit (Table 1 & Sec. 2.2). The new results are reported in the table below and we have added this experiments in Sec 2.3 and Table 2 in the revision.
> > > >
> > > > | Method            | 3B In-domain | 3B OOD | 7B In-domain | 7B OOD |
> > > > |-------------------|-------------:|-------:|-------------:|-------:|
> > > > | Zero-shot         | 49.5         | 38.5   | 62.0         | 43.6   |
> > > > | SFT               | 96.4         | 32.5   | 95.8         | 34.2   |
> > > > | Thinking-RFT      | 93.2         | 31.3   | 94.3         | 35.3   |
> > > > | No-Thinking RFT   | 95.8         | 32.0   | 96.1         | 34.0   |
> > > >
> > > > These numbers empirically reveal **4 opposite** trends of training on filtered/unfiltered data:
> > > >
> > > > 1. **Inverted ranking of post-training methods.**
> > > >    On Explore-ToM, the apparent hierarchy is SFT ≥ No-Thinking-RFT ≥ Thinking-RFT (all ≈93–96%), which is the **reverse** of our shortcut-free setting where Thinking-RFT > SFT > No-Thinking-RFT across four datasets. This illustrates that shortcuts hinder a faithful comparison of post-training methods and the misleading ranking would cause practitioners to prefer the wrong post-training strategy for enhancing ToM.
> > > >
> > > > 2. **Scaling effects are erased.**
> > > >    On clean OpenToM, increasing model size from 3B → 7B yields consistent +5–6 point gains for both SFT and Thinking-RFT. Under shortcut training, 3B and 7B collapse to almost identical in-domain performance (≈93–96% for all methods), and 3B SFT even slightly surpasses 7B SFT (96.4% vs 95.8%). We hypothesize that this is because overfitting cheap heuristics do not require much capacity. This also means that when trained on shortcut data, additional capacity brings no benefit and most capacity is wasted.
> > > >
> > > > 3. **Generalization is actively harmed (negative transfer).**
> > > >    For both 3B and 7B, tuning on Explore_ToM boosts in-domain accuracy by +40–45 points over zero-shot (e.g., 49.5 → 96.4 for 3B SFT), yet failes to generalize: on Hi-ToM performance *drops* by around −6–9 points (e.g., 38.5 → 31–33 for 3B; 43.6 → 34–35 for 7B). This is also the **opposite** of our cross-dataset genearlization results in Table 6 where RFT successfully improve performance even cross-dataset. This shows that shortcut data inflates in-domain accuracy and only induces a false sense of ToM.
> > > >
> > > > 4. **Shortcut training teaches no genuine ToM reasoning.**
> > > >    Complementing the accuracy numbers, we also perform the LLM-as-judge analysis and report the result below (also in Appendix H Table 10). The result shows that reasoning traces from shortcut-trained models score especially low in logical coherence (1/10), much lower than the Zero-shot+CoT baseline. On the same metric, RFT model on shortcut free data scores >9 on logical coherence.
> > > >
> > > > | Method                      | LLM-Judge (LC / F / E) |
> > > > |-----------------------------|--------------------------|
> > > > | Zero-shot                   | 4.5 / 3.2 / 8.0          |
> > > > | Thinking-RFT (w/ shortcut) | 1.0 / 9.8 / 8.2          |
> > > >
> > > >
> > > > Importantly, the opposite trends observed in experiments using/not using short-cut data is a good illustration of the effectiveness of our filtering method.

---

> > > > > ### Author Response · Authors · 2025-11-22
> > > > > **Rebuttal Part 5**
> > > > >
> > > > > ---
> > > > >
> > > > > >W7 ... as well as (b) the SFT/RFT-style post-training, can help mitigate such spurious correlation exploitation, even on those benchmarks with shortcuts.
> > > > >
> > > > > **R7.2** This is a good question. As demonstrated by the bad generalization behavior and undesirable results above in R7.1, shortcut only prevents models from developing genuine reasoning (it is also well-known that ML models tend to find the easiest path [4]) and **neither** post-training method (SFT/RFT) can directly resolve metigate issue. We note that suppressing shortcut learning online is a very interesting question and definitely a **worthwhile future direction** to explore. However, we acknowledge that developing such algorihtm is **highly non-trivial** and outside the scope of this work, hence we leave it for future work to explore. Our main contribution in this work lies in that we (i) raises the **shortcut issue** and demonstrate its harm to developing genuine ToM capability — potentially preventing a **misleading** research direction (Sec. 2.1 & 2.3), (ii) brings to the community’s attention the need to **rethink ToM benchmark making and evaluation** (Sec. 2.2), and (iii) demonstrate that on shortcut-free, high-quality data, it is entirely possible to **directly learn specialized ToM reasoning skills** (even for mid-sized models) via reinforcement-learning-based post-training (Sec. 3 & 4).
> > > > >
> > > > > ---
> > > > >
> > > > >
> > > > > **References**
> > > > >
> > > > > [1] ToMATO: Verbalizing the Mental States of Role-Playing LLMs for Benchmarking Theory of Mind
> > > > >
> > > > > [2] Explore Theory of Mind: Program-guided adversarial data generation for theory of mind reasoning
> > > > >
> > > > > [3] HI-TOM: A Benchmark for Evaluating Higher-Order Theory of Mind Reasoning in Large Language Models
> > > > >
> > > > > [4] Shortcut Learning in Deep Learning

---

> > > > > > ### Comment · Reviewer_MqzA · 2025-11-26
> > > > > >
> > > > > > I thank the authors for the comprehensive, insightful response. The new comparison with SimToM and AutoTom, as well as the results on the Hi-TOM and other cross-dataset evaluations, significantly strengthen the empirical contribution. I also appreciate the in-depth analysis of the spurious correlation, and the findings validate the motivation for removing shortcuts in the post-training dataset.
> > > > > >
> > > > > > Based on the revised manuscripts, I have decided to raise my rating to 6.

---

> > > > > > > ### Author Response · Authors · 2025-11-26
> > > > > > > **Thank you for reviewing our paper**
> > > > > > >
> > > > > > > We sincerely thank you again for your thoughtful comments and concrete suggestions and for increasing the score. We are glad that our additional results addressed your concerns and your positive feedback serves as a significant source of encouragement for our team!

---

### Official Review · Reviewer_8As9 · 2025-10-30

**Soundness:** 2
**Presentation:** 1
**Contribution:** 2
**Rating:** 4
**Confidence:** 3

**Summary:**

This paper studies how RL–based post-training can enhance ToM abilities in large language models. The authors identify that existing ToM datasets often contain shortcut patterns that allow models to achieve high accuracy without genuine reasoning. They propose a framework to audit such shortcuts and select four datasets for further study. Using these datasets, the paper compares Reinforcement FT, Supervised Fine-Tuning, and zero-shot baselines across several ToM tasks. The results show that RFT consistently outperforms SFT, particularly in higher-order reasoning, unseen domains, and counterfactual settings. Further analysis indicates that explicit reasoning steps and reinforcement learning jointly improve ToM robustness and generalization.

**Strengths:**

- Original framing of the ToM shortcut issue.

- The paper evaluates models on narrative, conversational, and multimodal ToM tasks.

- By connecting ToM, with RL techniques, the paper bridges human mental reasoning and computational learning frameworks in a creative and interdisciplinary manner.

**Weaknesses:**

- The introduction needs revision. It's hard to quickly grasp what challenge the paper aims to address from the current content.
Since Table 2 is mentioned early, it might help to first provide clear examples in the introduction so readers can easily understand the problem.

- The authors should double-check the references to figures and tables to ensure that all mentions correctly correspond to the intended visual content.

- It seems the authors did not re-implement or compare with previous ToM training methods; their comparison is limited to RFT, SFT, and zero-shot.
For a more complete experimental design, prior ToM methods should also be retrained on the authors' cleaned datasets and compared directly with RFT.

- Line 422–423: The explanation is unclear.
The authors should describe in more detail how to interpret Figure 5, and explicitly explain what the example is doing.

- The claim that explicit reasoning improves performance is not very novel, many NLP works have shown this.
What exactly is its effect in this specific task?
The authors should further analyze the reasoning content itself, both good and bad cases.

**Questions:**

- Line 62–63: "More importantly, the trained model produces incoherent and illogical reasoning traces as shown in Table 2."
How can this be observed from Table 2?

- Line 159–164: "As shown in Table 2, RFT mixes 'Jack's' own observation with the 4th order query being asked, ignoring intermediate ToM."
How can this be seen from Table 2?

---

> ### Author Response · Authors · 2025-11-22
> **Rebuttal Part 1**
>
> Thank you for acknowledging our shortcut finding, and we are grateful that you found our work "...bridges human mental reasoning and computational learning frameworks in a creative and interdisciplinary manner". We address your concerns in detail below.
>
> > W1. The introduction needs revision. It's hard to quickly grasp what challenge the paper aims to address from the current content. Since Table 2 is mentioned early, it might help to first provide clear examples in the introduction so readers can easily understand the problem.
>
> **R1.** Thank you for the feedback and suggestion. In the revised version, we **(i)** clarified the core challenge early the introduction and **(ii)** added a concrete shortcut example (Figure 1) into the fourth paragraph so that readers can quickly see the problem our work addresses.
>
>
> **The challenge.** There are two main challanges:
>
> - **Internalizing ToM reasoning.** ToM reasoning is intrinsically difficult: it requires the model to reason across a long horizon while tracking different objects, characters, which current foundation models lack [1-3]. Reasoning-based methods have been successfully proposed and shown great improvement [1,4-6]; however, these approaches typically rely on intricate multi-step prompting framework [1,6,5] around a strong backbone (e.g., GPT-4o) [4,5], which increases inference-time complexity and makes deployment more challenging. Our goal is to see if such ToM-specific reasoning can be **directly learned** via RL (and if so, when and how), without the manual hurdles.
>
> - **Shortcut-prone dataset.** When trying to solve (i), we discovered this critical issue that surprisingly exists in many ToM datasets. We have performed detaile experiments in Sec. 2.2 to highlight the harm of shortcut in detail — most importantly, it prevents models from developing genuine reasoning (as ML models tend to find the easiest path[7]), which goes directly against the goal for (i) (learning true ToM reasoning).
>
> Given these two challenges our work first addresses the shortcut issue by proposing a lightweight yet efficient audit framework (Sec. 2.1), identifying and removing the shortcut-heavy ones (Sec 2.2), and then study the first challange of efficiently learning human-level ToM reasoning (Sec. 3-4). By doing so our work not only shows that it is possible to learn ToM-specific reasoning through RL but also raises to the attention of the community a critical issue that could mislead future research effort.
>
> ---
>
>
> > W2. The authors should double-check the references to figures and tables to ensure that all mentions correctly correspond to the intended visual content.
>
> **R2.** We apologize for the negligence and thank the reviewer for pointing it out. We have corrected the reference (Figure 1) in the updated PDF version.
>
>
> ---
>
> > W3. It seems the authors did not re-implement or compare with previous ToM training methods; their comparison is limited to RFT, SFT, and zero-shot. For a more complete experimental design, prior ToM methods should also be retrained on the authors' cleaned datasets and compared directly with RFT.
>
> **R3.** This is a valid point and we thank the reviewer for bringing it up. To address the concern, we implemented two popular ToM algorithms, **SimToM** (prompting-based) [8] and **AutoToM** (Bayesian scaffolding) [5], and evaluated them on OpenToM and ToMATO using 1) the same backbone as our models (Qwen2.5-7B-Instruct) and 2) our RFT post-trained model as backbone. We have also added them in the updated version (Table 3 & 4)
>
>
> The results are reported in the table below. The results show that although SimToM and AutoToM do improve compared to the zero-shot baseline, our Thinking-RFT is **substantially stronger** (**+3.3/12.4 vs +42.7 on OpenToM**, the same trend is also observed on ToMATO). This shows that RL post-training on shortcut-free ToM data yields much larger gains than inference-time prompting or Bayesian scaffolding alone.
>
>
> | Method                   | OpenToM | ToMATO | MMToM | MuMA-ToM |
> |--------------------------|---------|--------|-------|----------|
> | Zero-shot                | 46.4    | 67.5   | 45.0  | 43.3     |
> | SimToM                   | 49.7    | 69.3   | 50.6  | 49.6     |
> | AutoToM                  | 56.8    | 69.4   | 56.9  | 59.1     |
> | Thinking-RFT             | 89.1    | 90.0   | 83.3  | 81.1     |
>
>
>  We kindly note that most ToM methods, including the two evaluated, are inference-time algorithms that do not update model weights, whereas ours (RFT post-training) does, thus they operate in a different regime. Also note that the original AutoToM paper employs GPT-4o as the backbone, hence lower performance is expected when switched to the weaker Qwen-7B model.

---

> > ### Author Response · Authors · 2025-11-22
> > **Rebuttal Part 2**
> >
> > > W4. Line 422–423: The explanation is unclear. The authors should describe in more detail how to interpret Figure 5, and explicitly explain what the example is doing.
> >
> > **R4.** Figure 5 provides a qualitative comparison of how RFT changes the model’s attention during reasoning, relative to a zero-shot+CoT baseline. Along the y-axis we index the generated output tokens in the reasoning/answer, and along the x-axis we index the input sentences (each column corresponds to one sentence, labeled by its first word; tokens within a sentence are aggregated). Each row therefore shows which sentence the model focuses on when generating a particular output token, with lighter colors indicating higher attention.
> >
> > In this example the model reads a story about Matthew and Grant and must classify Matthew’s attitude (positive / neutral / negative). The correct label depends on two key sentences: (i) Matthew loves bananas and wants easy access to them, and (ii) the banana is moved away from him. In panel (a), the RFT model consistently assigns high attention to these two “anchor” sentences when producing the crucial parts of its chain-of-thought and final answer, forming clear vertical bands over the highlighted columns. In contrast, in panel (b), the zero-shot+CoT model’s attention is much more diffuse and often concentrates on irrelevant sentences about Grant’s behavior. This pattern indicates that, after RFT on shortcut-free data, the model’s reasoning becomes better aligned with the minimal causal cues in the narrative, which is the kind of grounding required for robust ToM reasoning.”
> >
> > We have updated Sec. 5.2 to reflect the change.
> >
> > ---
> >
> >
> > > W5. The claim that explicit reasoning improves performance is not very novel, many NLP works have shown this.
> >
> > **R5.1** We acknowledge this point. We respecfully clarify that our goal is not only to show if RFT helps incentivize ToM reasoning, but also to clarify where and how it helps for ToM post-training. Moreover, a concurrent study [9] that also examines ToM post-training reaches the opposite conclusion: it finds that SFT can outperform RFT, which implies that explicit reasoning is not necessary. Our results, together with the shortcut audit in Sec. 2.2, identify a key missing piece in that picture: many ToM benchmarks are shortcut-prone, so conclusions drawn from them about the usefulness of reasoning (and RFT) can be misleading.

---

> > > ### Author Response · Authors · 2025-11-22
> > > **Rebuttal Part 3**
> > >
> > > > W5. What exactly is its effect in this specific task? The authors should further analyze the reasoning content itself, both good and bad cases.
> > >
> > > **R5.2** This is a good point. Unlike typical math or symbolic reasoning, real-world ToM involves multiple agents, long narratives, and evolving interactions, where the model must (i) filter out irrelevant details and focus on a small set of causal cues and (ii) continually track who knows what, when (agents, objects, and their mental states). Prior work has tried to inject these capabilities via Bayesian inverse planning or multi-round scaffolding [1,5]. Our contribution is to show that RFT with explicit human-like reasoning suffices to equip a base LM with these ToM-specific skills directly in its weights.
> > >
> > > Concretely, we offerred qualitative analysis in Sec. 5.2 and have added quantitative analysis in Sec. 5.1 and over the reasoning content. The example shown in Figure 5 shows that RFT successfullly helps to locate the cues (please refer to R4 above). To confirm this observation, we added a small experiment where we hired 3 PhD stuents to annote a 200 example set with anchor cues and found that RFT model finds them 89% of the time (vs. 35 for Zero-shot).
> > >
> > > To make these qualitative differences more explicit, we further added **two experiments/analyses in the revision (Sec. 5.1, Tab. 9)**: (i) **Reasoning Trace + Zero-shot**, where we provide the \texttt{<think>} trace from the RFT model (with the final answer removed) as a “hint” to the base model; good reasoning traces should in theory improve performance; (ii) **LLM-as-judge**, where a stronger LLM rates traces along logical consistency (LC), faithfulness to the story (F), and efficiency (E). The results are shown below
> > >
> > >
> > > | Method                        | OpenToM Acc | OpenToM LLM-Judge (LC/F/E) | ToMATO Acc | ToMATO LLM-Judge (LC/F/E) |
> > > |------------------------------|-------------|------------------------------|-------------|-----------------------------|
> > > | Zero-shot                    | 46.4        | 4.3 / 2.2 / 8.0              | 67.5        | 5.6 / 4.2 / 7.6             |
> > > | Thinking-RFT                 | 89.1        | 9.1 / 9.9 / 6.5              | 90.0        | 9.2 / 10.0 / 7.0            |
> > > | Zero-shot + RFT Reasoning Trace | 74.7        | –                            | 82.0        | –                           |
> > >
> > > As shown in Table above, feeding only the RFT traces to the frozen base model boosts accuracy from 46.4 → 74.7 on OpenToM and 67.5 → 82.0 on ToMATO (+28.3/+14.5), indicating that the RFT-generated chains-of-thought **encode useful ToM structure** rather than being post-hoc rationalizations. The LLM-judge further confirms this: Thinking-RFT almost doubles LC/F scores compared to Zero-shot (e.g., 4.3/2.2 → 9.1/9.9 on OpenToM), with only a mild efficiency trade-off. This indicates that the Thinking-RFT's reasoning is **logically coherent and correct** and **sticks to the fact**.
> > >
> > > ---
> > >
> > > > Q1. Line 62–63: "More importantly, the trained model produces incoherent and illogical reasoning traces as shown in Table 2." How can this be observed from Table 2?
> > >
> > > **A1.** We have fixed this (should be Figure 1) in the updated version.
> > >
> > > > Q2. Line 159–164: "As shown in Table 2, RFT mixes 'Jack's' own observation with the 4th order query being asked, ignoring intermediate ToM." How can this be seen from Table 2?
> > >
> > > **A2.** We have fixed this (should be Figure 1) in the updated version.
> > >
> > > **References**
> > >
> > > [1] MMToM-QA: Multimodal Theory of Mind Question Answering
> > >
> > > [2] HI-TOM: A Benchmark for Evaluating Higher-Order Theory of Mind Reasoning in Large Language Models
> > >
> > > [3] Explore Theory of Mind: Program-guided adversarial data generation for theory of mind reasoning
> > >
> > > [4] Think twice: Perspective-taking improves large language models’ theory-of-mind capabilities
> > >
> > > [5] Autotom: Scaling model-based mental inference via automated agent modeling
> > >
> > > [6] Muma-tom: Multi-modal multi-agent theory of mind
> > >
> > > [7] Shortcut Learning in Deep Learning
> > >
> > > [8] Think Twice: Perspective-Taking Improves Large Language Models' Theory-of-Mind Capabilities
> > >
> > > [9] Do Theory of Mind Benchmarks Need Explicit Human-like Reasoning in Language Models?

---

> > > > ### Comment · Reviewer_8As9 · 2025-11-25
> > > >
> > > > Thank you for your response. I have raised my score (4->6).

---

> > > > > ### Author Response · Authors · 2025-11-26
> > > > > **Thank you for reviewing our paper**
> > > > >
> > > > > We sincerely thank you again for your careful review and valuable suggestions and for kindly increasing the rating. Your positive feedback is a huge encouragement to our team.

---

### Official Review · Reviewer_rr8F · 2025-11-01

**Soundness:** 2
**Presentation:** 2
**Contribution:** 2
**Rating:** 4
**Confidence:** 4

**Summary:**

“From Shortcuts to Reasoning” examines how reinforcement fine-tuning (RFT) can enhance Theory of Mind (ToM) reasoning in language models. The authors first audit existing ToM datasets and reveal that many contain shortcuts—spurious cues that let models perform well without genuine reasoning. To address this, they curate shortcut-free datasets and compare RFT with supervised fine-tuning (SFT). Experimental results show that RFT yields stronger performance, particularly on second-order, multimodal, and counterfactual reasoning tasks. The paper concludes that reinforcement learning promotes more robust, causally grounded reasoning—but only when the data itself demands true inference rather than pattern matching.

**Strengths:**

- The paper provides a thoughtful and systematic critique of current Theory of Mind  benchmarks. It convincingly shows that many datasets contain *shortcuts* — spurious correlations or superficial cues that allow models to achieve high scores without genuine reasoning. This diagnostic insight is valuable, as it highlights that performance on standard ToM tests often overestimates a model’s true inferential ability.
- Through detailed experiments and analyses, the paper explores where RL-trained models perform better. It examines multiple dimensions — reasoning depth (first- vs. second-order ToM), modality (text vs. multimodal), and robustness (counterfactual consistency, generalization to unseen contexts).

**Weaknesses:**

- While the paper thoroughly shows that RL outperforms SFT, much of it confirms a result that is already broadly recognized in the field — that RL tends to yield higher task-specific performance. What’s missing is a deeper analysis of **how** RL-trained models differ qualitatively from SFT models in their reasoning traces. The attention map analysis is only an **indirect indicator**, and without a direct comparison between RL and SFT attention patterns, it’s difficult to understand whether RL’s gains reflect genuinely improved reasoning or simply better optimization.
- A key claim of the paper is that RFT improves reasoning only on shortcut-free datasets, while shortcut-prone ones hinder learning. However, this is **not empirically demonstrated**—the authors audit shortcut datasets but never show RFT results on them. As a result, the claim remains **conceptually convincing but experimentally unverified**; direct comparisons would have provided stronger evidence for RFT’s limitations.

**Questions:**

- Where is the result for comparing RFT including the datasets with shortcuts?
- What does the attention map look like for SFT?

---

> ### Author Response · Authors · 2025-11-22
> **Rebuttal Part 1**
>
> Thank you for your valueable feedback. We are encouraged that you recongized the importance of the "shortcut" issue to this community and found our experiments and analysis detailed. We address your concerns point by point in detail below.
>
> > W1：While the paper thoroughly shows that RL outperforms SFT, much of it confirms a result that is already broadly recognized in the field — that RL tends to yield higher task-specific performance.
>
> **R1.1** While it is true that for many tasks RFT is indeed better than SFT, this is not yet clear for ToM at the moment. In fact, a recent work [1] reaches the **opposite** conclusion, showing that SFT is instead better than RFT for ToM on all 3 tasks/benchmarks evaluated (Hi-ToM, ExploreToM, and ToMi) and that reasoning is not necessary for achieving the best performance. We show that this is not the case by carefully analyzing the datasets (Sec. 2.1) and raising the critical “shortcut” issue (Sec. 2.2). In this sense, we believe our findings support the ToM and broader post-training communities in designing more reliable benchmarks and tuning datasets, and help to avoid a potentially misleading research direction.
>
> Moreover, our study goes beyond the basic result RFT>SFT. We further disentangle the factor of RL and reasoning by experimenting with No-Thinking-RFT. The result of Thinking-RFT > SFT > No-Thinking-RFT highlights that RFT’s success at ToM rely on the **combination** of “RL and Reasoning” (Sec. 5.1).
>
>
> > W1: What’s missing is a deeper analysis of how RL-trained models differ qualitatively from SFT models in their reasoning traces.
>
> **R1.2** We appreciate this suggestion and agree that understanding qualitative differences in reasoning is important. In our current setup, however, the SFT model is trained with label-only supervision (no CoT annotations, as noted in Sec. 3, Line 220-222), so it is not explicitly taught to produce stable reasoning traces. This design choice keeps the supervision budget **comparable** and reflects a realistic setting where collecting high-quality ToM reasoning traces (either manually or via a stronger teacher model) is **costly and often impractical**. We have updated the text to make it clear. Instead, we analyze how RFT changes reasoning behavior by comparing Thinking-RFT against Zero-shot+CoT (Sec. 5.2, Fig. 5).
>
> To make these qualitative differences more explicit, we further added **two experiments/analyses in the revision (Sec. 5.1, Tab. 9)**: (i) **Reasoning Trace + Zero-shot**, where we provide the \texttt{<think>} trace from the RFT model (with the final answer removed) as a “hint” to the base model; good reasoning traces should in theory improve performance; (ii) **LLM-as-judge**, where a stronger LLM rates traces along logical consistency (LC), faithfulness to the story (F), and efficiency (E). The results are shown below
>
>
> | Method                        | OpenToM Acc | OpenToM LLM-Judge (LC/F/E) | ToMATO Acc | ToMATO LLM-Judge (LC/F/E) |
> |------------------------------|-------------|------------------------------|-------------|-----------------------------|
> | Zero-shot                    | 46.4        | 4.3 / 2.2 / 8.0              | 67.5        | 5.6 / 4.2 / 7.6             |
> | Thinking-RFT                 | 89.1        | 9.1 / 9.9 / 6.5              | 90.0        | 9.2 / 10.0 / 7.0            |
> | Zero-shot + RFT Reasoning Trace | 74.7        | –                            | 82.0        | –                           |
>
> As shown in Table above, feeding only the RFT traces to the frozen base model boosts accuracy from 46.4 → 74.7 on OpenToM and 67.5 → 82.0 on ToMATO (+28.3/+14.5), indicating that the RFT-generated chains-of-thought **encode useful ToM structure** rather than being post-hoc rationalizations. The LLM-judge further confirms this: Thinking-RFT almost doubles LC/F scores compared to Zero-shot (e.g., 4.3/2.2 → 9.1/9.9 on OpenToM), with only a mild efficiency trade-off. This indicates that the Thinking-RFT's reasoning is **logically coherent and correct** and **sticks to the fact**.
>
> > W1.  The attention map analysis is only an indirect indicator, and without a direct comparison between RL and SFT attention patterns, it’s difficult to understand whether RL’s gains reflect genuinely improved reasoning or simply better optimization.
>
> **R1.3** We kindly note that we originally **already provided** the attention map for SFT for the same problem in Fig 7 in the Appendix K. The results show that Thinking-RFT repeatedly attends to the two human-identified “anchor” sentences that form the causal hinge (“He liked X” and “X was moved away from him”) while SFT instead show minimal focus or over-focus on later distractor sentences and fail to align with the annotated minimal cues.

---

> > ### Author Response · Authors · 2025-11-22
> > **Rebuttal Part 2**
> >
> > ----
> >
> >
> > > W2. A key claim of the paper is that RFT improves reasoning only on shortcut-free datasets, while shortcut-prone ones hinder learning. However, this is not empirically demonstrated—the authors audit shortcut datasets but never show RFT results on them. As a result, the claim remains conceptually convincing but experimentally unverified; direct comparisons would have provided stronger evidence for RFT’s limitations.
> >
> > **R2.** This is a very good point, we agree with the reviewer and greatly appreciate the feedback. In the revision, we added an explicit **shortcut-training experiment** (Sec. 2.3, Tab. 2): we fine-tune 3B and 7B models with SFT, Thinking-RFT, and No-Thinking-RFT on the shortcut-prone Explore-ToM dataset, and evaluate both in-domain (Explore_ToM) and out-of-domain on Hi-ToM. We note that both benchmarks are flagged as shortcut-heavy by our audit (Tab. 1, Sec. 2.2). The new results are reported in the table below
> >
> > | Method            | 3B In-domain | 3B OOD | 7B In-domain | 7B OOD |
> > |-------------------|-------------:|-------:|-------------:|-------:|
> > | Zero-shot         | 49.5         | 38.5   | 62.0         | 43.6   |
> > | SFT               | 96.4         | 32.5   | 95.8         | 34.2   |
> > | Thinking-RFT      | 93.2         | 31.3   | 94.3         | 35.3   |
> > | No-Thinking RFT   | 95.8         | 32.0   | 96.1         | 34.0   |
> >
> > These numbers empirically reveal **4 drawback** of shortcut-prone data:
> >
> > 1. **Inverted ranking of post-training methods.**
> >    On Explore-ToM, the apparent hierarchy is SFT ≥ No-Thinking-RFT ≥ Thinking-RFT (all ≈93–96%), which is the **reverse** of our shortcut-free setting where Thinking-RFT > SFT > No-Thinking-RFT across four datasets. This illustrates that shortcuts hinder a faithful comparison of post-training methods and the misleading ranking would cause practitioners to prefer the wrong post-training strategy for enhancing ToM.
> >
> > 2. **Scaling effects are erased.**
> >    On clean OpenToM, increasing model size from 3B → 7B yields consistent +5–6 point gains for both SFT and Thinking-RFT. Under shortcut training, 3B and 7B collapse to almost identical in-domain performance (≈93–96% for all methods), and 3B SFT even slightly surpasses 7B SFT (96.4% vs 95.8%). We hypothesize that this is because overfitting cheap heuristics do not require much capacity. This also means that when trained on shortcut data, additional capacity brings no benefit and most capacity is wasted.
> >
> > 3. **Generalization is actively harmed (negative transfer).**
> >    For both 3B and 7B, tuning on Explore_ToM boosts in-domain accuracy by +40–45 points over zero-shot (e.g., 49.5 → 96.4 for 3B SFT), yet failes to generalize: on Hi-ToM performance *drops* by around −6–9 points (e.g., 38.5 → 31–33 for 3B; 43.6 → 34–35 for 7B). This is also the **opposite** of our cross-dataset genearlization results in Table 6 where RFT successfully improve performance even cross-dataset. This shows that shortcut data inflates in-domain accuracy and only induces a false sense of ToM.
> >
> > 4. **Shortcut training teaches no genuine ToM reasoning.**
> >    Complementing the accuracy numbers, we also perform the LLM-as-judge analysis and report the result below (also in Appendix H Table 10). The result shows that reasoning traces from shortcut-trained models score especially low in logical coherence (1/10), much lower than the Zero-shot+CoT baseline. The faithfulness metric is high because the shortcuts themselves are facts within the text, hence learning on them wouldn't decrease its score. However, the problematic logical reasoning proves that training on shortcuts does not result in genuine ToM reasoning.
> >
> > | Method                      | LLM-Judge (LC / F / E) |
> > |-----------------------------|--------------------------|
> > | Zero-shot                   | 4.5 / 3.2 / 8.0          |
> > | Thinking-RFT (w/ shortcut) | 1.0 / 9.8 / 8.2          |
> >
> >
> > So to sum up, these new experiments support our conceptual argument with direct empirical evidence: shortcut-prone datasets not only fail to reveal the benefits of RFT, they can invert method rankings, erase scaling effects, and even induce negative transfer.
> >
> >
> > > Q1. Where is the result for comparing RFT including the datasets with shortcuts?
> >
> > **A1.** Please refer to **R2** above.
> >
> > > Q2. What does the attention map look like for SFT?
> >
> > **A2.** Please refer to **R1.3** above.
> >
> > **References**
> >
> > [1] Do Theory of Mind Benchmarks Need Explicit Human-like Reasoning in Language Models?
> >
> > [2] Explore Theory of Mind: Program-guided adversarial data generation for theory of mind reasoning
> >
> > [3] HI-TOM: A Benchmark for Evaluating Higher-Order Theory of Mind Reasoning in Large Language Models
> >
> > ---

---

> > > ### Comment · Reviewer_rr8F · 2025-11-25
> > >
> > > Thank you for all the experiments in the rebuttal! These results and arguments make the paper stronger and clearer. Please add them to the final version. I raise my score to 6.

---

> > > > ### Author Response · Authors · 2025-11-26
> > > > **Thank you for reviewing our paper**
> > > >
> > > > We sincerely thank you again for your insightful comments and suggestions and for kindly increasing your score. We have incorporated new results in the updated manuscript and will make sure to discuss all details thoroughly in the final version!

---

### Author Response · Authors · 2025-11-22
**Global Response**

We extend our thanks to all four reviewers for their time and valuable, constructive feedback; these helped us to think more deeply and are immensely helpful in making our paper better. We are encouraged that the reviewers recognized the importance of the shortcut issue (rr8F, 8As9, MqzA, c2tS) we raised, as well as the thorough, detailed experiments and analyses (rr8F, MqzA) we performed in learning ToM-reasoning. We have carefully reviewed all their comments and have made every effort to address their concerns. We summarize our key contributions and revisions in the updated paper below. Detailed responses are also provided to the reviewers individually.

## **Main contributions**
---
- **Identifying shortcut issues in ToM benchmarks.**
This work conducts a thoughtful and systematic critique of current Theory of Mind benchmarks and raise a critical shortcut issue (rr8F) which has not been systematically documented before. It highlights that training on such datasets encourage models to exploit superficial correlations rather than develop genuine ToM capabilities (c2tS). This diagnostic insight also highlights that performance on standard ToM tests often overestimates a model’s true inferential ability (rr8F) and is worth studying deeper (MqzA).

- **Learning ToM-specialized reasoning via RL-based post-training.** This work conducts thorough, detailed experiments to demonstrate that desired ToM reasoning capabilities can be directly learned via RL-based post-training on clean data (rr8F, MqzA). The experiments validate this claim from multiple dimensions — reasoning depth, modality, and robustness (rr8F); the mechanistic analyses further reveal the cause behind the superior performance of Thinking-RFT (c2tS). By connecting ToM with RL techniques, this work bridges human mental reasoning and computational learning frameworks in a creative and interdisciplinary manner (8As9).


- **Impact.** **(i)** This work raises the shortcut issue and demonstrate its harm to developing genuine ToM capability — this is a critical issue that could potentially severely **misguide both future research and practitioners** aiming to improve ToM reasoning. **(ii)** It brings to the community’s attention the need to **rethink ToM benchmark making and evaluation**. **(iii)** This work shows that on shortcut-free, high-quality data, it is entirely possible to **directly learn specialized ToM reasoning** skills (even for mid-sized models) via reinforcement-learning-based post-training, achieving SOTA performance — this offers a simple, deployment-friendly alternative to other reasoning-based methods that may require intricate inference-time prompting procedures.

## **Major Updates**
---
- We added training results on shortcut-prone data for all methods and both model sizes in  **Table 2**
- We provided detailed quantitative and qualitative analysis of these shortcut-training results in **Sec. 2.3**
- We revised introduction for more clarity on the challange and contributions in **Sec. 1**
- We added two more popular baselines in main experiments in **Table 3 & 4**
- We added cross-dataset generalization results and anaysis in **Table 6 & Sec. 4.3**
- We added two reasoning quality evaluations and analysis in **Table 9 & Sec. 5.1**
- We added quantitative analysis over attention heatmap in **Appendix. J.3**
- We modified explanation of Figure 5 with more clarity in **Sec. 5.3**
- We provided more explanation over related work in **Sec. 6**
- We added more examples of reasoning traces and attention visualizations in **Appendix. K**

---

### Meta-Review · Area_Chair_ohjj · 2026-01-06

**Summary:**

This paper argues that existing ToM benchmarks contain **shortcuts** and proposes a post-training framework that uses filtered datasets to mitigate these issues. While the reviewers acknowledged that the rebuttal made the experimental evaluation more comprehensive, I remain concerned about the validity of the paper’s central claim regarding the existence and impact of dataset shortcuts.

The paper acknowledges that benchmarks such as HiToM and ExploreToM remain valid for evaluation because models do not have access to ground truth answers at test time. These datasets implement standard false-belief tasks, comparable to the foundational Sally Anne test, and are central to ToM research. The **shortcut learning** identified in the paper appears more plausibly to stem from the training methodology than from inherent issues in the datasets themselves. In particular, the behavior is consistent with reward hacking induced by the use of GRPO with **sparse, final outcome rewards**. By rewarding only the final answer, the training process incentivizes the model to exploit spurious correlations rather than develop robust intermediate reasoning. The reported poor generalization further points to an inappropriate reward formulation rather than a fundamental limitation of these classic epistemic logic tasks. With process-level rewards, such as supervision over intermediate state traces, this behavior would likely be reduced. For these reasons, I recommend rejecting this paper.

**Reviewer Concerns:**

The rebuttal substantially expanded the experimental evaluation and resolved many reviewer-raised issues regarding baselines, cross-dataset generalization, and qualitative analysis.

**Reviewer Scores:**

Although all reviewers raised their scores after the rebuttal due to improved empirical coverage and clearer analysis, I believe their assessments would remain overly optimistic in light of the unresolved question about the validity of the paper’s core claim.

---

### Decision · Program_Chairs · 2026-01-26

Reject